# Spread and Molecular Characteristics of *Enterobacteriaceae* Carrying *fosA*-Like Genes from Farms in China

Xiaoxiao Zhang,[a] Mingxiang Ma,[a] Yumeng Cheng,[a] Yiqin Huang,[a] Yuxiao Tan,[a] Yunqiao Yang,[a] Yajing Qian,[a] Xin Zhong,[a] Yujie Lu,[a] Hongbin Si[a]

[a]College of Animal Science and Technology, State Key Laboratory for Conservation and Utilization of Subtropical Agro-bioresources, Guangxi University, Nanning, China

Xiaoxiao Zhang and Mingxiang Ma contributed equally to this work. Author order was determined by the corresponding author after negotiation.

**ABSTRACT** In this study, we aimed to investigate the occurrence and molecular characteristics of fosfomycin-resistant *Enterobacteriaceae* isolates from pig, chicken and pigeon farms in Guangxi Province of China. A total of 200 fosfomycin-resistant strains were obtained from food animals and their surrounding environments, with the *fosA*, *fosA3*, and *fosA7.5* genes being detected in 26% (52/200), 10% (20/200), and 5% (10/200), respectively. Surprisingly, three *fosA7.5*-producing *E. coli* isolates were found to be concomitant with *fosA3*. Most of the *fosA*-like-gene-positive isolates were multidrug-resistant strains and consistently possessed $bla_{CTX-M-1/CTX-M-9}$, *floR*, and $bla_{TEM}$ genes. Only *fosA3* was successfully transferred to the recipient strains, and the 29 *fosA3*-carrying transconjugants exhibited high-level resistance to fosfomycin (MIC ≥ 512 µg/mL). Multilocus sequence typing (MLST) combined with enterobacterial repetitive intergenic consensus-PCR (ERIC-PCR) analyses indicated that *fosA3* or *fosA7.5* genes were spread by horizontal transfer as well as via clonal transmission between *E. coli*. We used the PCR mapping method to explore the genetic contexts of *fosA*-like genes, and two representative strains (fEc.1 and fEcg99-1) were fully sequenced. Six different genetic structures surrounding *fosA3* were detected and one infrequent context was discovered among the conjugable *fosA3*-positive *E. coli* isolates. The five genetic environments of *fosA* were identified and found to be highly similar to the partial sequence of transposon Tn2921. Furthermore, whole-genome sequencing (WGS) results showed that *fosA7.5* was colocalized with *mcr-3*, $bla_{CMY-63}$, *sul3*, *tet*(A), *dfrA*, and a number of virulence-related factors on the same chromosomes of strains, and various insertion sequences (IS3/ISL3) were detected upstream or downstream of *fosA7.5*. The phylogenetic analysis revealed that both *fosA7.5*- and *fosA3*-carrying *E. coli* ST602 and *fosA7.5*-carrying *E. coli* ST2599 were closely related to *E. coli* isolates from humans, which may indicate that they pose a threat to human health.

**IMPORTANCE** Here, we report the widespread and complex genetic environments of *fosA*-like genes in animal-derived strains in China. The *fosA7.5* gene was identified in this study and was found to confer resistance to fosfomycin. The high prevalence of *fosA*-like genes in farms indicates that food animals serve as a potential reservoir for the resistance genes. This study also discovered that fosfomycin resistance genes were always associated with mobile elements, which would accelerate the transmission of *fosA*-like genes in strains. Importantly, *E. coli* ST602 and ST2599 carrying *fosA3* or *fosA7.5* from food animals had high similarity to *E. coli* isolates from humans, suggesting that *fosA*-like genes can be transmitted to humans through the food chain, thus posing a serious threat to public health. Therefore, the prevalence of *fosA*-like genes isolated from animals should be further monitored.

**KEYWORDS** food animals, *Enterobacteriaceae*, *fosA*-like genes, fosfomycin resistance, transmission, farms, fosfomycin, genetic environments

Address correspondence to Hongbin Si, shb2009@gxu.edu.cn.

The authors declare no conflict of interest.

The wide spread of multidrug-resistant (MDR) Gram-negative bacteria, such as extended-spectrum-$\beta$-lactamase (ESBL)-producing *Enterobacteriaceae* and carbapenem-resistant *Enterobacteriaceae* (CRE), has resulted in fewer options for clinical treatment. In this case, fosfomycin, an older antibiotic agent, has garnered renewed interest and is considered a first-line antibiotic to treat infections caused by carbapenem-resistant and polymyxin-resistant bacteria (1). However, as the use of fosfomycin increased, so did the widespread dissemination of fosfomycin-resistant isolates in some countries. It has already been reported that fosfomycin resistance is relatively severe in China, with the resistance rates ranging from 25% to 50% (2–4). However, fosfomycin is still effective against ESBL-producing *Enterobacteriaceae* such as *Salmonella*, *Escherichia coli*, and *Klebsiella pneumoniae* in Europe, the Americas, and Africa (5).

Resistance to fosfomycin is primarily mediated by the expression of fosfomycin-modifying enzymes (FosA, FosB, and FosC), whereas the FosA enzyme encoded by chromosomes or plasmids is the most common in Gram-negative bacteria. To date, more than 10 *fosA*-like genes (*fosA1* to *fosA10*) have been identified, of which *fosA3* encodes the primary mechanism leading to fosfomycin resistance of *E. coli* and *K. pneumoniae* in China (6–8). Presently, *fosA3* is widely distributed among *Enterobacteriaceae* strains isolated from pets, pigs, chickens, and cows, as well as humans, although fosfomycin is not approved for use in animals in China (9–11). Furthermore, the coexistence of *fosA3* with other antibiotic resistance determinants ($bla_{CTX-M}$, $bla_{TEM}$, and *floR*) on plasmids has resulted in the emergence of fosfomycin-resistant strains in various countries around the world (12).

Previous research discovered that *fosA7* is mainly found on the chromosomes of *Salmonella* from various sources (human, cattle, sheep, and environment) (13). Subsequently, this gene was detected in different countries (14–16). In 2020, a study reported that the FosA identified in *Escherichia coli* differed from FosA7 protein, which was first reported in *Salmonella*, and its encoding gene was named *fosA7.5*$^{Q86E}$ (17). At present, *fosA7.5* mainly exists in *E. coli*, and three variants of *fosA7.5* were discovered, of which *fosA7.5*$^{Q86E}$ and *fosA7.5*$^{WT}$ demonstrated resistance to fosfomycin. Moreover, the fosfomycin resistance gene *fosA* described in *Serratia marcescens* in 1980, which was the first *fosA*-like gene (namely, *fosA1*), also could confer high-level resistance to fosfomycin (18). However, limited information is available regarding the prevalence of *fosA* and *fosA7* among *Enterobacteriaceae* isolated from food animals, and no study has ever reported that *fosA3* and *fosA7.5* are coharbored in a single *E. coli* strain.

As a result, the strains containing *fosA*, *fosA3*, or *fosA7.5* from food animals and their environments were analyzed in this study to better understand their resistance phenotypes, plasmid replicon typing, genetic environments, and transmission characteristics. It provides a scientific foundation for future efforts to prevent the spread of fosfomycin resistance genes at the human-animal-environment interface.

## RESULTS

**Identification of fosfomycin resistance determinants and coexisting resistance genes.** In this study, a total of 200 fosfomycin-resistant *Enterobacteriaceae* isolates were obtained from the samples. Among these 200 strains, 82 were positive for *fosA*-like genes, and they came from chicken feces ($n = 36$), pig feces ($n = 6$), sewage from pig farms ($n = 3$), pig lungs ($n = 4$), pig nose ($n = 4$), pig mouth ($n = 6$), soil from pig farms ($n = 4$), soil from chicken farms ($n = 4$), pig anus ($n = 1$), pigeon ($n = 12$), and shells of chicken eggs ($n = 2$). Among the 82 *fosA*-like-gene-positive isolates, including 52 *fosA3*-positive *E. coli* isolates (26%; 52/200), 10 *fosA7.5*-positive *E. coli* isolates from pigeons (10%; 10/200), and 20 *fosA*-positive isolates (*Enterobacter cloacae* ($n = 10$), *Escherichia hormaechei* ($n = 7$) and *Escherichia asburiae* ($n = 3$) isolates) were also identified by 16S rRNA sequencing. Importantly, in the 10 *fosA7.5*-harboring *E. coli* isolates, three strains (KPg84, fEc.1, and ECg85) coharbored both *fosA7.5* and *fosA3*. However, *fosC2* and other fosfomycin resistance genes were not detected. Detailed information for the 82 strains is shown in Table S1 in the supplemental material.

**TABLE 1** Characterization of 29 conjugable *fosA3*-positive *E. coli* isolates

| Strains | Context of *fosA3*[a] | Resistance profile[b] | Resistance genes |
|---|---|---|---|
| EC27 | V | FFC, CHL, TET, CIP, FOS | *fosA3*, *bla*$_{CTX-M-9}$, *bla*$_{TEM}$, *rmtB*, *floR* |
| EC28 | V | FFC, CHL, TET, CIP, FOS | *fosA3*, *bla*$_{CTX-M-9}$, *bla*$_{TEM}$, *floR* |
| EC29 | I | CAZ, FFC, CHL, TET, CIP, FOS | *fosA3*, *bla*$_{CTX-M-1}$, *bla*$_{TEM}$, *floR* |
| EC30 | I | CAZ, FFC, CHL, TET, TGC, FOS | *fosA3*, *bla*$_{CTX-M-1}$, *bla*$_{TEM}$, *floR* |
| EC31 | II | CAZ, FFC, CHL, TET, FOS | *fosA3*, *bla*$_{CTX-M-1}$, *bla*$_{TEM}$, *floR* |
| EC32 | II | CAZ, FFC, CHL, TET, CIP, AMK, COL, FOS | *fosA3*, *bla*$_{CTX-M-1}$, *bla*$_{TEM}$, *floR* |
| EC33 | IV | CAZ, FFC, CHL, TET, CIP, FOS | *fosA3*, *bla*$_{CTX-M-1}$, *bla*$_{TEM}$, *floR* |
| EC34 | V | FFC, CHL, TET, CIP, FOS | *fosA3*, *bla*$_{CTX-M-9}$, *bla*$_{TEM}$, *floR* |
| EC35 | VI | FFC, CHL, TET, CIP, FOS | *fosA3*, *bla*$_{TEM}$, *floR* |
| EC36 | VI | CAZ, FFC, CHL, TET, CIP, AMK, FOS | *fosA3*, *bla*$_{CTX-M-1}$, *bla*$_{TEM}$, *rmtB*, *floR* |
| EC37 | I | CAZ, FFC, CHL, TET, CIP, FOS | *fosA3*, *bla*$_{CTX-M-1}$, *bla*$_{TEM}$, *floR* |
| EC38 | VI | FFC, TET, FOS | *fosA3*, *floR* |
| EC39 | II | FFC, CHL, TET, CIP, TGC, FOS | *fosA3*, *bla*$_{TEM}$, *floR* |
| EC40 | IV | CAZ, FFC, CHL, TET, CIP, FOS | *fosA3*, *bla*$_{CTX-M-9}$, *floR* |
| EC41 | VI | CAZ, FFC, CHL, TET, CIP, TGC, FOS | *fosA3*, *bla*$_{CTX-M-1}$, *bla*$_{TEM}$, *floR* |
| EC42 | V | FFC, CHL, TET, CIP, FOS | *fosA3*, *bla*$_{CTX-M-9}$, *bla*$_{TEM}$, *floR* |
| EC43 | IV | CAZ, FFC, CHL, TET, CIP, COL, FOS | *fosA3*, *bla*$_{CTX-M-1}$, *bla*$_{CTX-M-9}$, *bla*$_{TEM}$, *floR* |
| EC44 | II | CAZ, FFC, CHL, TET, CIP, FOS | *fosA3*, *bla*$_{CTX-M-1}$, *bla*$_{TEM}$, *floR* |
| EC45 | I | CAZ, FFC, CHL, TET, CIP, FOS | *fosA3*, *bla*$_{CTX-M-1}$, *floR* |
| EC46 | VI | CAZ, FFC, CHL, TET, CIP, TGC, FOS | *fosA3*, *bla*$_{CTX-M-1}$, *rmtB*, *floR* |
| EC47 | II | CAZ, FFC, CHL, TET, CIP, FOS | *fosA3*, *bla*$_{CTX-M-1}$, *bla*$_{TEM}$, *floR* |
| EC48 | II | CAZ, FFC, CHL, TET, CIP, FOS | *fosA3*, *bla*$_{CTX-M-9}$, *bla*$_{TEM}$, *rmtB*, *floR* |
| EC49 | VI | CAZ, FFC, CHL, TET, CIP, FOS | *fosA3*, *bla*$_{CTX-M-1}$, *bla*$_{TEM}$, *floR* |
| EC50 | V | FFC, CHL, TET, CIP, FOS | *fosA3*, *bla*$_{CTX-M-9}$, *bla*$_{TEM}$, *floR* |
| EC51 | VI | FFC, CHL, TET, CIP, TGC FOS | *fosA3*, *bla*$_{CTX-M-9}$, *bla*$_{TEM}$, *floR* |
| EC52 | I | FFC, CHL, TET, AMK, TGC, FOS | *fosA3*, *bla*$_{CTX-M-1}$, *bla*$_{TEM}$, *rmtB*, *floR* |
| Kpg84 | / | CAZ, FFC, CHL, TET, CIP, FOS | *fosA3*, *fosA7.5*, *bla*$_{CTX-M-1}$, *floR*, *bla*$_{TEM}$ |
| fEc.1 | III | CAZ, FFC, CHL, TET, CIP, FOS | *fosA3*, *fosA7.5*, *bla*$_{CTX-M-1}$, *floR*, *bla*$_{TEM}$ |
| ECg85 | / | CAZ, FFC, CHL, TET, CIP, FOS | *fosA3*, *fosA7.5*, *bla*$_{CTX-M-1}$, *floR*, *bla*$_{TEM}$ |

[a]/, the genetic environment of fosA3 was not detected.
[b]CAZ, ceftazidime; FFC, florfenicol; CHL, chloramphenicol; TET, tetracycline; CIP, ciprofloxacin; AMK, amikacin; COL, colistin; TGC, tigecycline; MEM, meropenem; FOS, fosfomycin.

The *fosA/fosA3/fosA7.5*-carrying *Enterobacteriaceae* isolates were also tested for the presence of other significant antibiotic resistance genes (ARGs). Screening for resistance genes confirmed that 40 of the 52 *fosA3*-positive *E. coli* isolates carried *bla*$_{CTX}$-like resistance genes, and strain EC43 contained two different *bla*$_{CTX-M}$ genes, including *bla*$_{CTX-M-1G}$ and *bla*$_{CTX-M-9G}$. In addition, 9 and 35 isolates harbored *rmtB* and *bla*$_{TEM}$ genes, respectively, and all *fosA3*-carrying isolates were positive for *floR*. As a result, we identified the following gene combinations for *fosA3*: *fosA3-bla*$_{CTX-M-1}$-*bla*$_{TEM}$-*rmtB*-*floR* (n = 7), *fosA3-bla*$_{CTX-M-9}$-*bla*$_{TEM}$-*rmtB*-*floR* (n = 2) *fosA3-bla*$_{CTX-M-9}$-*bla*$_{TEM}$-*floR* (n = 5), *fosA3-bla*$_{CTX-M-1}$-*bla*$_{TEM}$-*floR* (n = 15), *fosA3-bla*$_{CTX-M-1}$-*bla*$_{CTX-M-9}$-*bla*$_{TEM}$-*floR* (n = 1), *fosA3-floR* (n = 6), *fosA3-bla*$_{TEM}$-*floR* (n = 6), *fosA3-bla*$_{CTX-M-1}$-*floR* (n = 1), *fosA3-bla*$_{CTX-M-9}$-*floR* (n = 8), and *fosA3-bla*$_{CTX-M-9}$-*rmtB*-*floR* (n = 1) (Table 1; also, see Fig. 1). Except for strains ECg29 and EC315, all other *fosA7.5*-positive *E. coli* isolates carried *bla*$_{CTX-M}$, *bla*$_{TEM}$, and *floR* genes, and the most frequent gene profile was *fosA3/fosA7.5-bla*$_{CTX-M-1/CTX-M-9}$-*floR*-*bla*$_{TEM}$ (n = 8) (Table 2). However, most of the 20 *fosA*-positive strains showed a single-gene profile, and only one and four strains carried *bla*$_{NDM}$- and *bla*$_{CTX}$-like resistance genes, respectively (Table 2). The rates of *floR*, *bla*$_{TEM}$, and *rmtB* genes were relatively low, at 20% (4/20), 10% (2/20), and 5% (1/20).

**Detection of antimicrobial resistance patterns.** In this study, 82 *Enterobacteriaceae* isolates containing *fosA*-like genes showed different degrees of resistance to 10 antimicrobial agents (Fig. 2A and B). Susceptibility testing indicated that all 82 strains were resistant to fosfomycin (100%; 82/82). These fosfomycin-resistant isolates also showed resistance to other antibiotics, such as ceftazidime (58.54%; 48/82), florfenicol (95.12%; 78/82), and chloramphenicol (85.37%; 70/82), tetracycline (90.24%; 74/82) and ciprofloxacin (73.17%; 60/82), and the resistance rates were all above 55%. However, only one and 10 strains were resistant to meropenem (1.22%) and amikacin (12.20%),

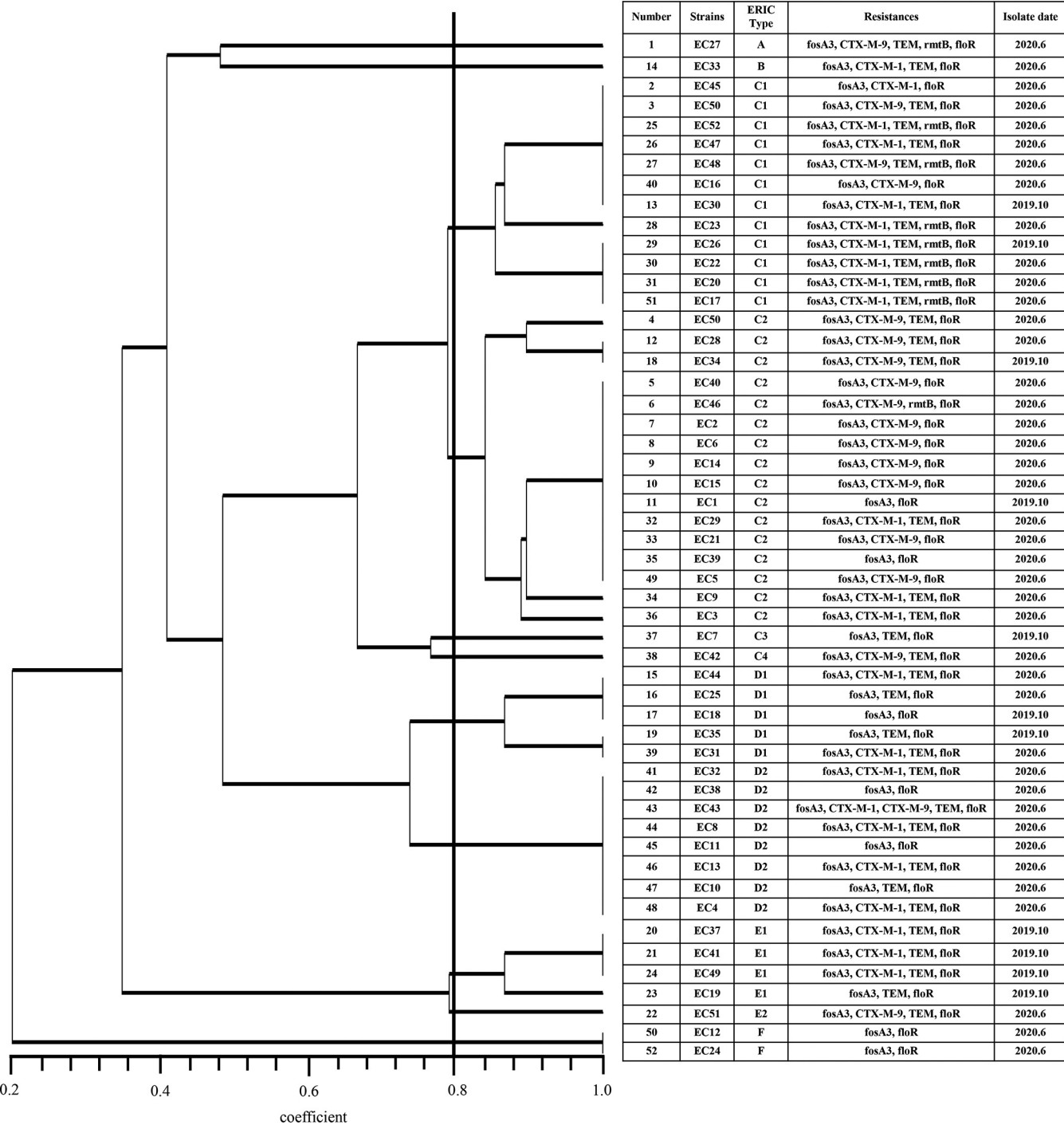

| Number | Strains | ERIC Type | Resistances | Isolate date |
|---|---|---|---|---|
| 1 | EC27 | A | fosA3, CTX-M-9, TEM, rmtB, floR | 2020.6 |
| 14 | EC33 | B | fosA3, CTX-M-1, TEM, floR | 2020.6 |
| 2 | EC45 | C1 | fosA3, CTX-M-1, floR | 2020.6 |
| 3 | EC50 | C1 | fosA3, CTX-M-9, TEM, floR | 2020.6 |
| 25 | EC52 | C1 | fosA3, CTX-M-1, TEM, rmtB, floR | 2020.6 |
| 26 | EC47 | C1 | fosA3, CTX-M-1, TEM, floR | 2020.6 |
| 27 | EC48 | C1 | fosA3, CTX-M-9, TEM, rmtB, floR | 2020.6 |
| 40 | EC16 | C1 | fosA3, CTX-M-9, floR | 2020.6 |
| 13 | EC30 | C1 | fosA3, CTX-M-1, TEM, floR | 2019.10 |
| 28 | EC23 | C1 | fosA3, CTX-M-1, TEM, rmtB, floR | 2020.6 |
| 29 | EC26 | C1 | fosA3, CTX-M-1, TEM, rmtB, floR | 2019.10 |
| 30 | EC22 | C1 | fosA3, CTX-M-1, TEM, rmtB, floR | 2020.6 |
| 31 | EC20 | C1 | fosA3, CTX-M-1, TEM, rmtB, floR | 2020.6 |
| 51 | EC17 | C1 | fosA3, CTX-M-1, TEM, rmtB, floR | 2020.6 |
| 4 | EC50 | C2 | fosA3, CTX-M-9, TEM, floR | 2020.6 |
| 12 | EC28 | C2 | fosA3, CTX-M-9, TEM, floR | 2020.6 |
| 18 | EC34 | C2 | fosA3, CTX-M-9, TEM, floR | 2019.10 |
| 5 | EC40 | C2 | fosA3, CTX-M-9, floR | 2020.6 |
| 6 | EC46 | C2 | fosA3, CTX-M-9, rmtB, floR | 2020.6 |
| 7 | EC2 | C2 | fosA3, CTX-M-9, floR | 2020.6 |
| 8 | EC6 | C2 | fosA3, CTX-M-9, floR | 2020.6 |
| 9 | EC14 | C2 | fosA3, CTX-M-9, floR | 2020.6 |
| 10 | EC15 | C2 | fosA3, CTX-M-9, floR | 2020.6 |
| 11 | EC1 | C2 | fosA3, floR | 2019.10 |
| 32 | EC29 | C2 | fosA3, CTX-M-1, TEM, floR | 2020.6 |
| 33 | EC21 | C2 | fosA3, CTX-M-9, floR | 2020.6 |
| 35 | EC39 | C2 | fosA3, floR | 2020.6 |
| 49 | EC5 | C2 | fosA3, CTX-M-9, floR | 2020.6 |
| 34 | EC9 | C2 | fosA3, CTX-M-1, TEM, floR | 2020.6 |
| 36 | EC3 | C2 | fosA3, CTX-M-1, TEM, floR | 2020.6 |
| 37 | EC7 | C3 | fosA3, TEM, floR | 2019.10 |
| 38 | EC42 | C4 | fosA3, CTX-M-9, TEM, floR | 2020.6 |
| 15 | EC44 | D1 | fosA3, CTX-M-1, TEM, floR | 2020.6 |
| 16 | EC25 | D1 | fosA3, TEM, floR | 2020.6 |
| 17 | EC18 | D1 | fosA3, floR | 2019.10 |
| 19 | EC35 | D1 | fosA3, TEM, floR | 2019.10 |
| 39 | EC31 | D1 | fosA3, CTX-M-1, TEM, floR | 2020.6 |
| 41 | EC32 | D2 | fosA3, CTX-M-1, TEM, floR | 2020.6 |
| 42 | EC38 | D2 | fosA3, floR | 2020.6 |
| 43 | EC43 | D2 | fosA3, CTX-M-1, CTX-M-9, TEM, floR | 2020.6 |
| 44 | EC8 | D2 | fosA3, CTX-M-1, TEM, floR | 2020.6 |
| 45 | EC11 | D2 | fosA3, floR | 2020.6 |
| 46 | EC13 | D2 | fosA3, CTX-M-1, TEM, floR | 2020.6 |
| 47 | EC10 | D2 | fosA3, TEM, floR | 2020.6 |
| 48 | EC4 | D2 | fosA3, CTX-M-1, TEM, floR | 2020.6 |
| 20 | EC37 | E1 | fosA3, CTX-M-1, TEM, floR | 2019.10 |
| 21 | EC41 | E1 | fosA3, CTX-M-1, TEM, floR | 2019.10 |
| 24 | EC49 | E1 | fosA3, CTX-M-1, TEM, floR | 2019.10 |
| 23 | EC19 | E1 | fosA3, TEM, floR | 2019.10 |
| 22 | EC51 | E2 | fosA3, CTX-M-9, TEM, floR | 2020.6 |
| 50 | EC12 | F | fosA3, floR | 2020.6 |
| 52 | EC24 | F | fosA3, floR | 2020.6 |

**FIG 1** ERIC-PCR profiles of 52 *fosA3*-positive *E. coli* isolates.

respectively. It was also found that 16 strains (19.51%) exhibited intermediate resistance to amikacin (MIC = 4 $\mu$g/mL). Furthermore, several isolates were resistant to colistin (17.07%; 4/62) and tigecycline (20.97%; 14/82), with MICs at or above 2 $\mu$g/mL, and the resistant strains were mostly detected among *fosA*-positive isolates (Fig. 2C). Except for one strain that was only resistant to two antibiotics (fosfomycin and florfenicol), all 81 strains carrying *fosA*-like genes were multidrug-resistant strains (resistant to at least 3 classes of agents). According to the findings, 79 strains (98.75%) were resistant to 4 or more antibiotics, and six strains were resistant to all 8 antibiotics (Fig. 2D). The MICs of 82 strains are listed in Table S2 and S3.

**TABLE 2** Characterization of 10 *fosA7.5*-positive isolates and 20 *fosA*-positive isolates

| Strain | Resistance profile[a] | Resistance gene(s) |
|---|---|---|
| Kpg84 | CAZ, FFC, CHL, TET, CIP, FOS | *fosA3, fosA7.5, bla*$_{CTX-M-1}$*, floR, bla*$_{TEM}$ |
| fEc.1 | CAZ, FFC, CHL, TET, CIP, FOS | *fosA3, fosA7.5, bla*$_{CTX-M-1}$*, floR, bla*$_{TEM}$ |
| ECg85 | CAZ, FFC, CHL, TET, CIP, FOS | *fosA3, fosA7.5, bla*$_{CTX-M-1}$*, floR, bla*$_{TEM}$ |
| fEcg991 | CAZ, FFC, CHL, TET, CIP, FOS | *fosA7.5, bla*$_{CTX-M-9}$*, floR, bla*$_{TEM}$ |
| ECg29 | CAZ, FFC, CHL, TET, CIP, FOS | *fosA7.5, floR, bla*$_{TEM}$ |
| ECg931 | CAZ, FFC, CHL, TET, CIP, FOS | *fosA7.5, bla*$_{CTX-M-1}$*, floR, bla*$_{TEM}$ |
| ECg932 | CAZ, FFC, CHL, TET, CIP, FOS | *fosA7.5, bla*$_{CTX-M-1}$*, bla*$_{CTX-M-9}$*, floR* |
| ECg91 | CAZ, FFC, CHL, TET, CIP, AMK, FOS | *fosA7.5, bla*$_{CTX-M-1}$*, bla*$_{CTX-M-9}$*, floR* |
| ECg933 | CAZ, FFC, CHL, TET, CIP, FOS | *fosA7.5, bla*$_{CTX-M-1}$*, bla*$_{CTX-M-9}$*, floR* |
| EC315 | FFC, FOS | *fosA7.5, floR, bla*$_{TEM}$ |
| 20E.1 | FFC, FOS, TET | *fosA* |
| 20E.2 | FFC, TGC, FOS, TET | *fosA* |
| EC2088 | FFC, CHL, TET, TGC, FOS | *fosA, floR* |
| 20E.4 | FFC, CHL, TET, TGC, FOS | *fosA, floR* |
| 20E.5 | FFC, TET, TGC, FOS | *fosA, bla*$_{TEM}$ |
| 20E.6 | FFC, COL, TGC, FOS | *fosA* |
| 20E.7 | TET, COL, TGC, FOS | *fosA* |
| 20E.8 | TET, COL, TGC, FOS | *fosA* |
| 20E.9 | FFC, TET, COL, TGC, FOS | *fosA, bla*$_{CTX-M-9}$ |
| EC2098 | FFC, CHL, TET, CIP, FOS | *fosA, floR* |
| 20E.11 | FFC, CHL, TET, FOS | *fosA* |
| KP20117 | FFC, CHL, TET, COL, TGC, FOS | *fosA* |
| 20E.13 | CAZ, FFC, CHL, TET, CIP, COL, TGC, FOS | *fosA, bla*$_{CTX-M-9}$ |
| 20E.14 | CAZ, FFC, CHL, TET, CIP, COL, TGC, FOS | *fosA, bla*$_{CTX-M-9}$ |
| 20E.15 | CAZ, FFC, CHL, TET, CIP, AMK, FOS | *fosA, rmtB* |
| 20E.16 | CAZ, TET, CIP, COL, FOS | *fosA* |
| 20E.17 | CAZ, FFC, CHL, TET, COL, TGC, FOS | *fosA, bla*$_{TEM}$*, floR, bla*$_{CTX-M-1}$ |
| 20E.18 | CAZ, TET, MEM, FOS | *fosA, bla*$_{NDM}$ |
| EC1928 | FFC, CHL, TET, CIP, TGC, FOS | *fosA* |
| 20E.20 | FFC, COL, TGC, FOS | *fosA* |

[a]CAZ, ceftazidime; FFC, florfenicol; CHL, chloramphenicol; TET, tetracycline; CIP, ciprofloxacin; AMK, amikacin; COL, colistin; TGC, tigecycline; MEM, meropenem; FOS, fosfomycin.

**Conjugation experiments and plasmid analysis.** Among the 82 *fosA/fosA3/fosA7.5*-harboring isolates, 29 (35.37%; 29/82) were able to successfully transfer the fosfomycin resistance phenotype to *E. coli* recipient strain C600, and all transconjugants carried *fosA3*. For the three *E. coli* isolates coharboring both *fosA7.5* and *fosA*, only *fosA3* was successfully transferred from three donors to the recipient. Moreover, no *fosA* or *fosA7.5* transconjugants were acquired, indicating that these genes may be located on chromosomes or non-conjugative plasmids of strains. The MICs of 7 antimicrobial agents for 29 transconjugants are listed in Table 3, all of which were resistant to fosfomycin (MIC > 512 $\mu$g/mL). Furthermore, the 29 transconjugants showed resistance to ceftazidime (62.07%; 18/29), florfenicol (86.21%; 25/29), chloramphenicol (86.21%; 25/29), tetracycline (68.97%; 20/29), ciprofloxacin (17.24%; 5/29), and amikacin (10.34%; 3/29). It was found that 24 (82.76%) *fosA3*-carrying transconjugants were resistant to more than 4 antibiotics (Fig. S3). Enterobacterial repetitive intergenic consensus-PCR (ERIC-PCR) indicated that the bands of conjugants were consistent with *E. coli* C600, while showing differences with the donors (Fig. S4).

Except for three strains that carried both *fosA3* and *fosA7.5*, 26 conjugable *fosA3*-positive *E. coli* isolates included a total of 8 different plasmid replicon types, including Inc (I1, FIA, FIB, FII, K, HI1, HI2, N), and all of the strains contained 2 or more plasmid replicons (Table 4). The corresponding transconjugants also acquired different plasmid replicons; only IncFIB replicons were detected in two transconjugants (EC47-T and EC48-T), indicating that *fosA3* was located on Inc(FIB) plasmids. The results showed that multiple plasmids were transferred horizontally with the *fosA3* plasmids. Different from *fosA3*-positive isolates, seven plasmid replicons were detected in 10 *fosA7.5*-positive *E. coli* isolates (Table S4), including Inc (F$_{repB}$, FIB, FII, I1, K, and Y). F$_{repB}$ was discovered in all *fosA7.5*-positive *E. coli* strains, while IncY was found in six strains.

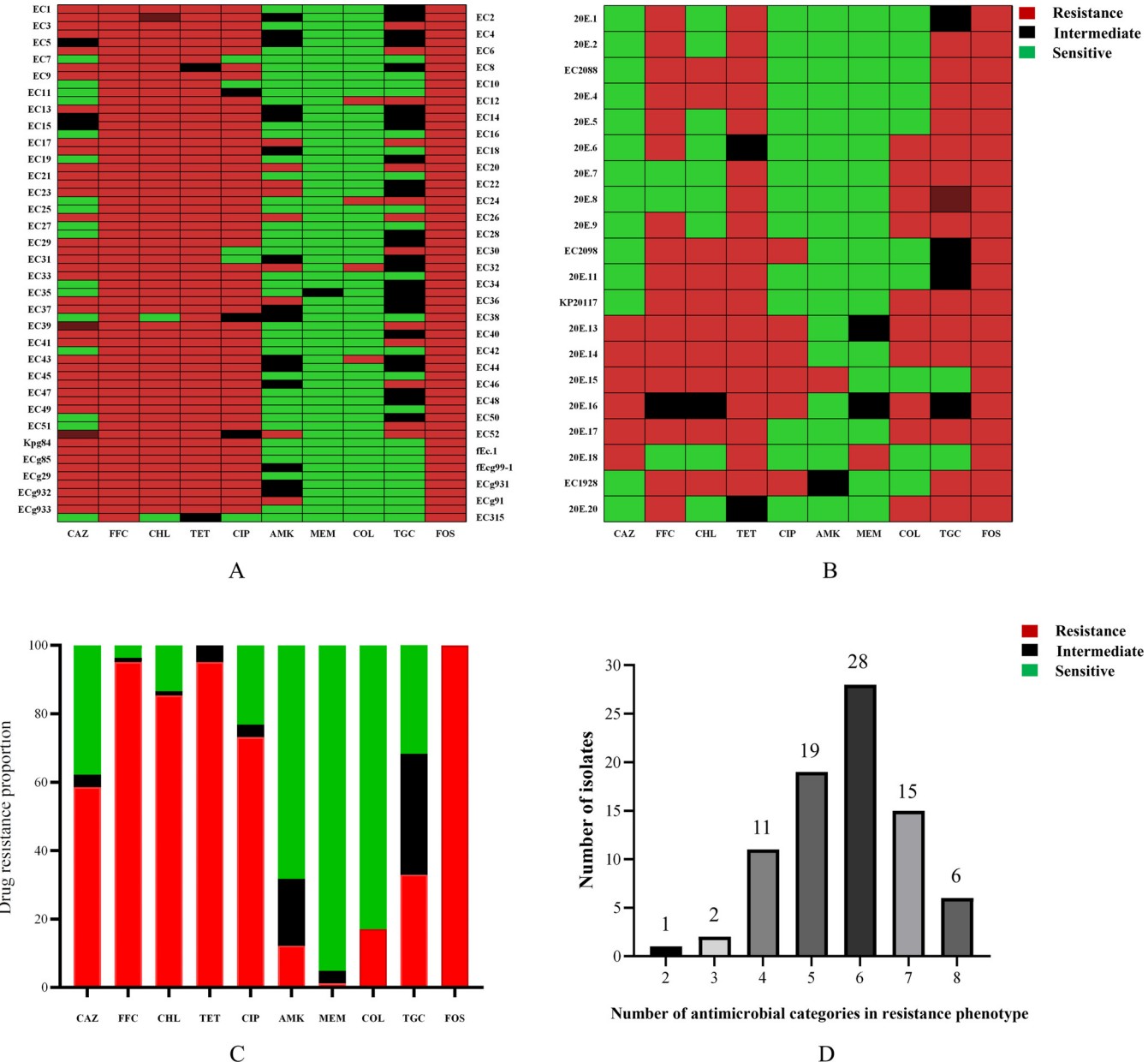

**FIG 2** Analysis of the susceptibility results of 82 *Enterobacteriaceae* isolates with fosfomycin resistance for 13 antibiotics. (A and B) Drug resistance spectrum; (C) drug resistance proportion; (D) numbers of isolates with given numbers of antimicrobial categories in the resistance phenotypes.

**Strain typing (ERIC-PCR and MLST).** The genomic diversity analysis of 52 *fosA3*-positive strains and 10 *fosA7.5*-positive strains was analyzed by using the ERIC-PCR fingerprinting method. Among the 52 *fosA3*-positive *E. coli* isolates, the number of amplified bands ranged from 3 to 10, with sizes of 100 bp to 2,000 bp, and the genetic similarity was 20% to 100%. These isolates were divided into 6 main clusters (A to F) and 11 ERIC types, of which cluster C (C1 to C4) was the dominant genotype (59.62%; 31/52), and most of the strains in cluster C were derived from animal feces. Clusters A and B had the fewest strains, with only one strain in each cluster. The remaining five clusters (D to F) contained 13 (25%), 5 (9.62%) and 1 (3.85%) isolates, respectively (Fig. 1). MLST revealed a new sequence type (ST) and 15 known STs for the 29 conjugable *fosA3*-positive *E. coli* isolates, in which ST115 was the most common (*n* = 5), followed by ST156 (*n* = 4), ST7069 (*n* = 3), ST117 (*n* = 3), ST1196 (*n* = 2), ST23 (*n* = 2). Other STs were ST5229, ST683, ST202, ST224, ST410, ST1148, ST602, ST1468, and ST48, and each ST had one isolate. The two known STs (ST410

**TABLE 3** MICs of 10 antimicrobial agents for the 29 *fosA3* transconjugants

| Strain | MIC (μg/mL) of[a]: | | | | | | | |
|---|---|---|---|---|---|---|---|---|
| | CAZ | FFC | CHL | TET | CIP | AMK | RIF | FOS |
| EC27-T | <1 | 128 | 32 | 2 | <1 | <1 | >1,000 | >512 |
| EC28-T | <1 | 256 | 128 | 64 | <1 | 2 | >1,000 | >512 |
| EC29-T | 8 | 2 | 2 | 2 | <1 | <1 | >1,000 | >512 |
| EC30-T | 16 | 512 | 128 | 128 | <1 | 2 | >1,000 | >512 |
| EC31-T | 4 | 256 | 64 | 32 | <1 | <1 | >1,000 | >512 |
| EC32-T | 16 | 4 | 2 | 2 | <1 | 2 | >1,000 | >512 |
| EC33-T | 8 | 256 | 256 | 32 | <1 | <1 | 125 | >512 |
| EC34-T | <1 | 128 | 64 | 64 | <1 | <1 | >1,000 | >512 |
| EC35-T | <1 | 256 | 128 | 128 | 8 | 2 | >1,000 | >512 |
| EC36-T | 8 | 256 | 64 | 64 | <1 | <1 | >1,000 | >512 |
| EC37-T | 16 | 512 | 128 | 256 | <1 | <1 | >1,000 | >512 |
| EC38-T | <1 | 512 | 256 | 256 | <1 | <1 | >1,000 | >512 |
| EC39-T | 8 | 512 | 256 | 128 | 512 | <1 | 1,000 | >512 |
| EC40-T | 16 | 512 | 256 | 256 | 128 | 2 | >1,000 | >512 |
| EC41-T | 8 | 256 | 64 | 128 | <1 | 2 | >1,000 | >512 |
| EC42-T | <1 | 128 | 64 | 32 | <1 | 2 | >1,000 | >512 |
| EC43-T | 16 | 2 | 2 | 2 | <1 | <1 | >1,000 | >512 |
| EC44-T | 8 | 4 | 2 | 64 | <1 | <1 | >1,000 | >512 |
| EC45-T | 8 | 256 | 128 | 16 | <1 | <1 | >1,000 | >512 |
| EC46-T | 8 | 256 | 64 | 512 | 64 | 2 | >1,000 | >512 |
| EC47-T | <1 | 8 | 64 | 4 | <1 | 8 | >1,000 | >512 |
| EC48-T | <1 | 8 | 64 | 4 | <1 | 16 | >1,000 | >512 |
| EC49-T | 16 | 512 | 256 | 256 | <1 | <1 | >1,000 | >512 |
| EC50-T | <1 | 128 | 64 | 64 | <1 | <1 | >1,000 | >512 |
| EC51-T | <1 | 256 | 128 | 64 | <1 | <1 | >1,000 | >512 |
| EC52-T | 8 | 128 | 128 | 64 | 2 | 8 | >1,000 | >512 |
| fEc.1-T | 8 | 256 | 128 | 2 | <1 | <1 | >1,000 | >512 |
| Kpg84-T | 8 | 256 | 64 | <1 | <1 | <1 | >1,000 | >512 |
| ECg85-T | 8 | 256 | 128 | 2 | <1 | <1 | >1,000 | >512 |
| C600 | <1 | <1 | 2 | 2 | <1 | <1 | >1,000 | <1 |

[a]CAZ, ceftazidime; FFC, florfenicol; CHL, chloramphenicol; TET, tetracycline; CIP, ciprofloxacin; AMK, amikacin; RIF, rifampicin; FOS, fosfomycin.

and ST23) belong to clonal complex 23 (CC23) and had only one difference in their *purA* alleles. The allele profiles of STs are provided in Table 5.

MLST analysis showed that the 10 *fosA7.5*-positive *E. coli* isolates belonged to 4 STs (one ST1468, one ST602, six ST2599, and one ST7051), in which ST2599 was predominant. Analysis of the ERIC-PCR profiles showed that there were a total of 3 unique clusters (A, B, and C) and 5 ERIC types within 10 *fosA7.5*-carrying *E. coli* isolates. Except for strain ECg931, other ST2599 and ST7051 strains belonged to cluster C. Also, the three *E. coli* isolates carrying both *fosA3* and *fosA7.5* were classified as cluster A (Fig. 3). ERIC-PCR combined with MLST analyses indicated that the *fosA*-like genes were spread by horizontal transfer as well as via clonal transmission between *Enterobacteriaceae* isolates in the farms. The allele profiles of STs are provided in Table 5.

**Genetic background of *fosA* in *E. cloacae* and *E. hormaechei*.** For the 17 *fosA*-positive isolates, four types of genetic contexts were identified by PCR mapping, all of which shared > 99% similarity with partial sequences in *E. cloacae* strain ECNIH5 (CP009854) and *S. marcescens* transposon Tn*2921* (FJ829469). The most common were type I (*n* = 7) and type III (*n* = 3), while the others were type II (*n* = 1) and type IV (*n* = 1). In all four types, a 247-bp length of amplicon in the upstream region of *fosA* was identical to the truncated tryptophan tRNA synthetase gene in *E. cloacae* strain ECNIH5 and transposon Tn*2921*. In the downstream region of *fosA*, we found four amplicons with lengths of 957 bp, 894/1,045 bp, 1,203 bp, and 576 bp encoding the LacI family transcriptional regulator, sugar-binding cellulose-like protein, MFS sugar transporter, and restriction endonuclease, respectively, which were similar to part of the sequence in transposon Tn*2921*. In type I, only a

**TABLE 4** Plasmid replicons of the 29 *fosA3*-positive *E. coli* and their transconjugants

| Strain | Plasmid types | Transconjugant | Plasmid type(s) |
|---|---|---|---|
| EC27 | HI2, FIB, FII, K | EC27-T | FIB, FII |
| EC28 | FIB, FII, K | EC28-T | FIB, FII |
| EC29 | I1, FIA, FIB, FII, K | EC29-T | I1, FIB, FII |
| EC30 | I1, FIB, FII, K | EC30-T | FIB, FII |
| EC31 | FIB, FII, K | EC31-T | FIB, FII |
| EC32 | HI1, HI2, N, FIB, FII | EC32-T | N, FIB, FII |
| EC33 | HI1, FIB, FII, K | EC33-T | FIB, FII |
| EC34 | FIB, FII, K | EC34-T | FIB, FII |
| EC35 | FIB, FII | EC35-T | FIB, FII |
| EC36 | N, FIB, B, FII, K | EC36-T | N, FIB, FII |
| EC37 | FIB, FII, K | EC37-T | FIB, FII |
| EC38 | HI1, FIB, FII | EC38-T | FIB, FII |
| EC39 | FIB, FII, K | EC39-T | FIB, FII |
| EC40 | FIB, FII, K | EC40-T | FIB, FII |
| EC41 | FIB, FII, K | EC41-T | FIB, FII |
| EC42 | HI2, FIB, FII, K | EC42-T | FIB, FII |
| EC43 | HI1, HI2, N, FIB, FII | EC43-T | N, FIB, FII |
| EC44 | FIB, FII, K | EC44-T | FIB, FII |
| EC45 | HI1, FIB, FII, K | EC45-T | FIB, FII |
| EC46 | FIB, FII, K | EC46-T | FIB, FII |
| EC47 | I1, N, FIB, B, FII, K | EC47-T | FIB |
| EC48 | I1, N, FIB, B, FII, K | EC48-T | FIB |
| EC49 | FIB, FII, K | EC49-T | FIB, FII |
| EC50 | FIB, FII, K | EC50-T | FIB, FII |
| EC51 | FIB, FII, K | EC51-T | FIB, FII |
| EC52 | I1, FIB, FII, K | EC52-T | FIB, FII |
| Kpg84 | F$_{repB}$, FIB, FII, I1, K | Kpg84-T | I1, FIB, FII |
| fEc.1 | F$_{repB}$, FIB, FII, I1, K | Ecg87-T | I1, FIB, FII |
| ECg85 | F$_{repB}$, FIB, I1, FII, K | Kpg85-T | I1, FIB, FII |
| fEcg99-1 | F$_{repB}$, FIB, I1, Y, FII, K | None | None |

582-bp length of amplicon encoding the LacI family transcriptional regulator was confirmed in the downstream region of *fosA* (Fig. 4).

**Genetic background of *fosA3* in *E. coli* isolates.** PCR mapping was used to determine the regions adjacent to *fosA3* in 26 conjugable *fosA3*-positive *E. coli* isolates.

**TABLE 5** The ST types and of conjugable *fosA3*-positive *E. coli* and *fosA7.5*-carrying isolates

| No. of allele genes | | | | | | | | |
|---|---|---|---|---|---|---|---|---|
| *adk* | *fumC* | *gyrB* | *icd* | *mdh* | *purA* | *recA* | ST[a] | Strain(s) |
| 6 | 6 | 33 | 26 | 11 | 8 | 2 | 1196 | EC27, EC42 |
| 6 | 4 | 14 | 16 | 24 | 8 | 14 | 115 | EC28, EC34, EC35, EC44, EC50 |
| 112 | 11 | 5 | 12 | 8 | 8 | 6 | 7069 | EC37, EC41, EC49 |
| 6 | 11 | 4 | 8 | 8 | 8 | 2 | 48 | EC52 |
| 43 | 41 | 15 | 18 | 11 | 7 | 44 | 5229 | EC29 |
| 20 | 45 | 41 | 43 | 5 | 32 | 2 | 117 | EC32, EC38, EC43 |
| 6 | 4 | 12 | 1 | 20 | 13 | 7 | 23 | EC47, EC48 |
| 6 | 4 | 127 | 16 | 24 | 8 | 6 | 683 | EC30 |
| 6 | 4 | 12 | 1 | 20 | 18 | 7 | 410 | EC36 |
| 6 | 4 | 33 | 16 | 11 | 8 | 6 | 224 | EC33 |
| 6 | 95 | 3 | 18 | 11 | 7 | 14 | 1148 | EC45 |
| 6 | 29 | 32 | 16 | 11 | 8 | 44 | 156 | EC40, EC46, EC39, EC51 |
| 64 | 11 | 5 | 8 | 5 | 8 | 2 | 202 | EC31 |
| 6 | 19 | 33 | 26 | 11 | 8 | 6 | 602 | fEC.1 |
| 6 | 6 | 153 | 26 | 11 | 8 | 6 | 1468 | ECg85 |
| 267 | 6 | 5 | 26 | 9 | 13 | 98 | 2599 | fEcg99-1, ECg931, ECg933, ECg932, ECg91, ECg29 |
| 6 | 19 | 33 | 26 | 11 | 8 | 98 | NA | Kpg84 |
| 653 | 19 | 270 | 26 | 11 | 8 | 7 | 7051 | EC315 |

[a]NA, no ST type of the strain has been obtained.

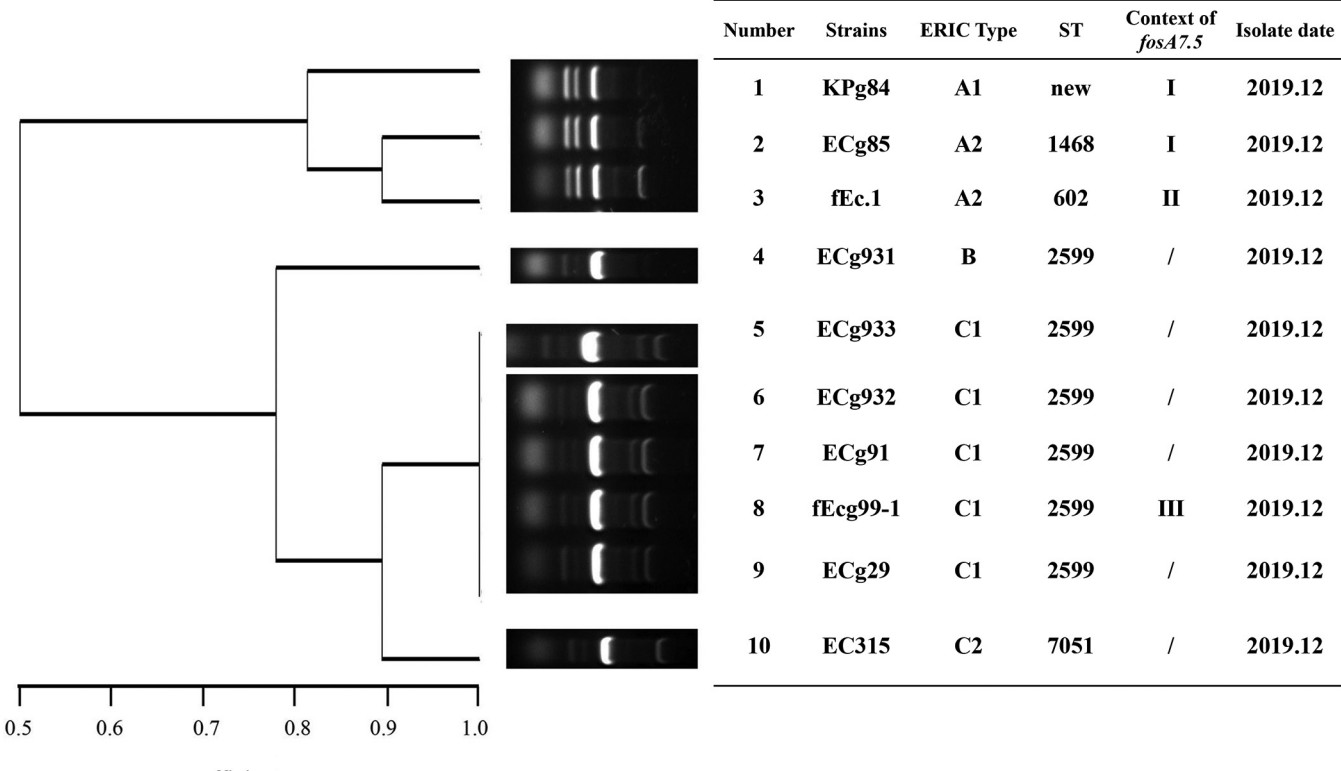

| Number | Strains | ERIC Type | ST | Context of *fosA7.5* | Isolate date |
|--------|---------|-----------|-----|----------------------|--------------|
| 1 | KPg84 | A1 | new | I | 2019.12 |
| 2 | ECg85 | A2 | 1468 | I | 2019.12 |
| 3 | fEc.1 | A2 | 602 | II | 2019.12 |
| 4 | ECg931 | B | 2599 | / | 2019.12 |
| 5 | ECg933 | C1 | 2599 | / | 2019.12 |
| 6 | ECg932 | C1 | 2599 | / | 2019.12 |
| 7 | ECg91 | C1 | 2599 | / | 2019.12 |
| 8 | fEcg99-1 | C1 | 2599 | III | 2019.12 |
| 9 | ECg29 | C1 | 2599 | / | 2019.12 |
| 10 | EC315 | C2 | 7051 | / | 2019.12 |

**FIG 3** ERIC-PCR profiles of 10 *fosA7.5*-positive *E. coli* isolates.

Five different genetic contexts were identified, including type I (*n* = 4), type II (*n* = 6), type IV (*n* = 3), type V (*n* = 6), and type VI (*n* = 7) (Fig. 5). The *fosA3* gene was flanked by two IS*26* elements oriented in the opposite direction in 20 isolates. An IS*26* element was found to be located on downstream of *fosA3* in all isolates, and the lengths of the spacer regions between the 3′ end of *fosA3* and the IS*26* gene were variable (2,377 bp, 1,823 bp, and 707 bp). In type I, II, IV, and VI structures, the IS*26* element was located 385 bp upstream of *fosA3*. In addition, the extended-spec-

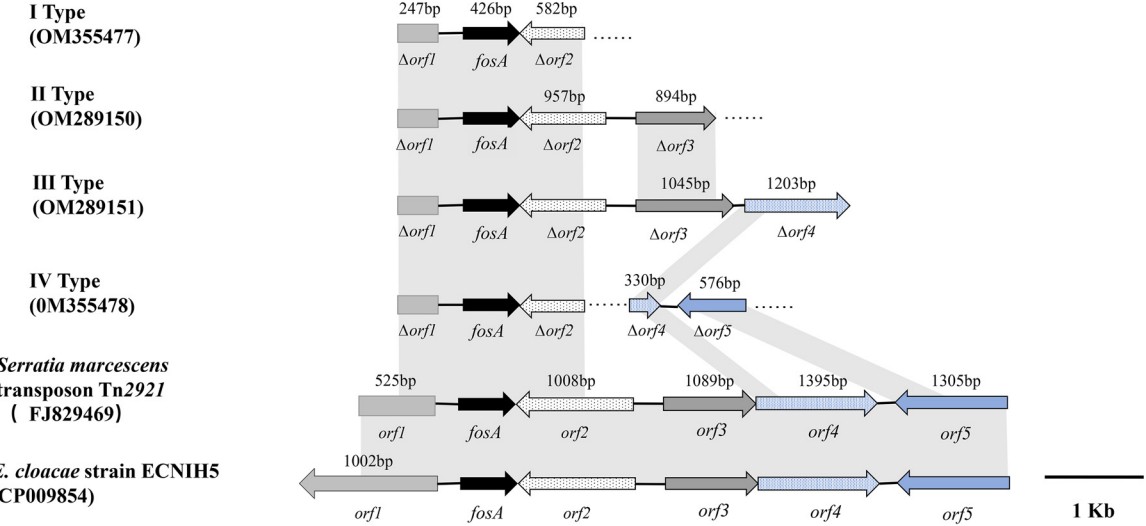

**FIG 4** Genetic contexts of *fosA* in *E. cloacae* and *E. hormaechei*. *orf1*, *orf2*, *orf3*, *orf4*, and *orf5* encode part of the tryptophan tRNA synthetase, LacI family transcriptional regulator, glycosyl hydrolase family 2, MFS sugar transporter, and a restriction endonuclease. Shaded boxes between sequences indicate homologous regions (>90% sequence identity).

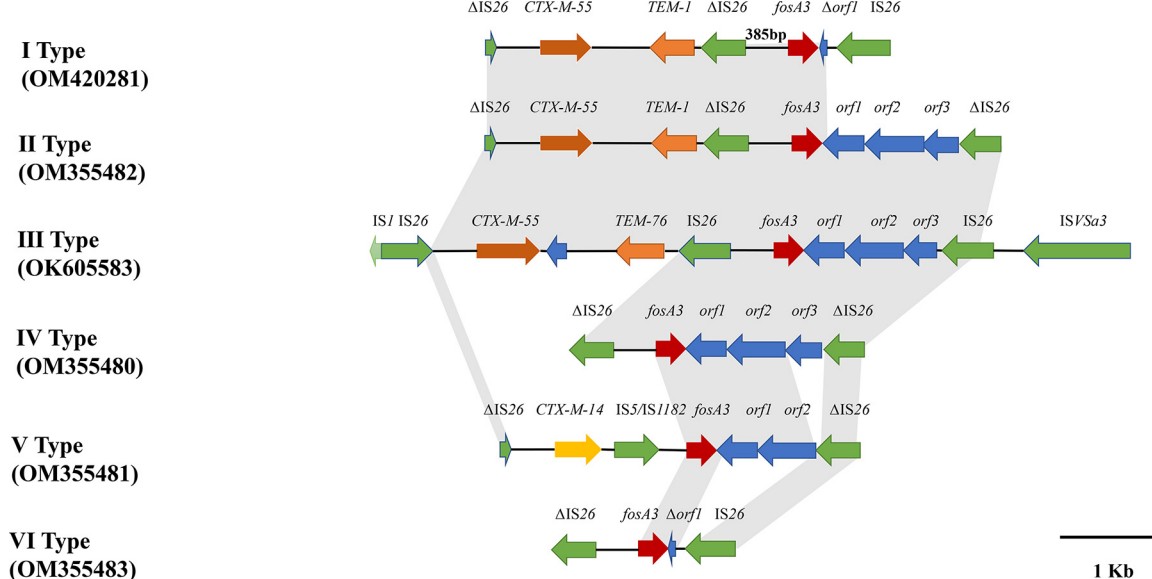

**FIG 5** Genetic context of *fosA3* in *E. coli*. Arrows indicate the directions of transcription of the genes, and different genes are shown in different colors. Shaded boxes between sequences indicate homologous regions (>90% sequence identity). *orf1*, *orf2*, and *orf3* encode a hypothetical protein, a CadC-like protein, and a truncated TetR family transcriptional regulator.

trum β-lactamase (ESBL) gene $bla_{CTX-M-55}$ was frequently located upstream of *fosA3* in two genomic contexts (type I and type II), and a truncated IS*26* transposase determinant was identified upstream of $bla_{CTX-M-55}$. The type V (*n* = 6) structure was from $bla_{CTX-M-14}$-positive isolates; it was found that the IS*26* element upstream of *fosA3* was replaced by the ΔIS*26*-$bla_{CTX-M-14}$-IS*5*/IS*1182*-*fosA3* structure, which was similar to that on plasmid on LWY24 (MT318677.1, chicken, *E. coli*) (Fig. 5).

One representative pigeon-derived *E. coli* isolate (fEC.1) carrying *fosA3* and *fosA7.5* was analyzed by whole-genome sequencing (WGS) and was identified as ST602. The *fosA3*-harboring plasmid was named pfEC.1-3 (OK605583), with a size of 78,319 bp. The plasmid belonged to the IncFII incompatibility group and contained a variable region responsible for *fosA3*. The two structures, IS*1*-IS*26*-*orf*-$bla_{CTX-M-55}$-*orf*-$bla_{TEM}$-76-IS*26*-*fosA3* and *fosA3*-*orf1*-*orf2*-*orf3*-IS*26*-IS*Vsa3*/IS*91*-*floR*-*aph(3')-Ia*, were located upstream and downstream of *fosA3*, respectively, and were named type III. A BLAST search for pfEC.1-3 revealed highly homology (>90%) to six other known IncFII plasmids deposited in the GenBank database, which were p14E509-2FII (MN822125.1; China; human), pCREC-591_2 (CP024823.1; South Korea; human), pCTX-M-55_005237 (CP026576.2; China; human), pHNGD4P177 (MG197492.1; China; pig), pHNMC02 (MG197489.1; China; chicken) and pT224A (MW298658.1; Canada; dairy cow). All plasmids had backbone genes associated with IncFII plasmid replication (*repA1/A2*), conjugative transfer and the type IV secretion system (T4SS) (*tra* and *trb*), separation (*parM*), and maintenance of genetic stability (*stbA*) (Fig. 6). However, the variations between these plasmids resulted from insertion sequences (IS*26*, IS*4*, and IS*91*), integrase (IntI), and resistance [*floR* and *aph(3')-Ia*] genes around the *fosA3* gene (Fig. 7).

**Phylogenetic analysis of fEC.1.** Phylogenetic analysis was performed by using WGS information available in the GenBank database for ST602 fEC.1 and 28 *E. coli* isolates (ST602, *n* = 23; ST5498, *n* = 1; ST unknown, *n* = 4) from clinical samples from different sources, including humans, animals, and plants, and one *E. coli* isolate of unknown origin. The phylogenetic analysis by core genome MLST (cgMLST) revealed that ST602, ST5498 and 4 unknown STs were classified into the same lineage, indicating clonal spread of these strains. Isolate fEC.1 from this study was most closely related to two ST602 *E. coli* isolates, 13KWH46 (CP019250) and HB_Coli0 (CP020933), collected from a patient and chicken feces, which both carried *fosA7*, *mdf(A)*, *floR*, *aph(3')-IIb*, *aph*

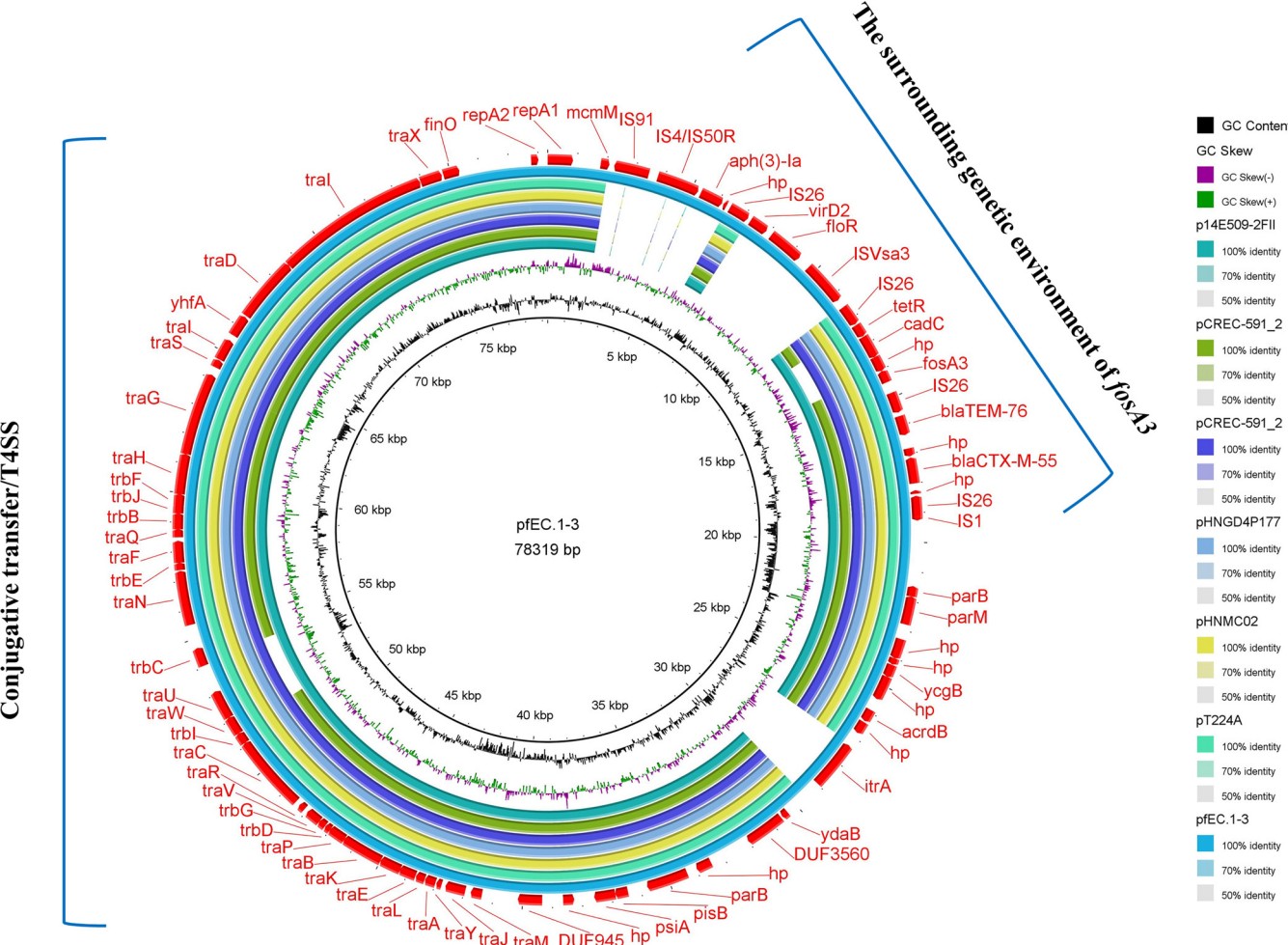

**FIG 6** Comparative genomics analysis of IncFII plasmids carrying *fosA3*, the external ring represents the annotation of pfEC.1-3.

*(6')-Id*, *sul2*, *tet*(A), and *tet*(B) genes. Importantly, these strains were also abundant in distribution, including some countries in Asia, Africa, North America, and Europe, suggesting that an ST602 *E. coli* isolate is spreading across host species and continents. In addition, the majority of the strains carried fosfomycin resistance genes, including *fosA3* and *fosA7*, and showed a multiresistance gene profile (Fig. 8).

**Genetic background of *fosA7.5* in *E. coli* isolates.** In addition to strain fEC.1, a *fosA7.5*-carrying *E. coli* isolate (fEcg99-1) from a pigeon was also completely sequenced to analyze the genetic environment of *fosA7.5*. The MLST scheme revealed that fEcg99-1 belonged to sequence type ST2599. This study identified 3 different genetic contexts associated with the *fosA7.5* gene, designated type I to III (Fig. 9). WGS revealed that *fosA7.5* was located on the chromosome in strains fEc.1 (type II) and fEcg99-1 (type III). In all three types, a gene sequence containing *orf2*, *orf3*, *orf4*, and *orf5* was found downstream of *fosA7.5* that encoded HNH endonuclease, sialate-*O*-acetylesterase, sialic acid-induced transmembrane protein YjhT (NanM), and *N*-acetylneuraminic acid outer membrane channel protein (NanC), respectively. According to comparative genomic analysis, the three structures (type I to III) were highly similar to part of *E. coli* AH01 (CP055251.1). However, the difference was that the *orf3* sequence lengths in the three types were 573 bp, 1,008 bp, and 759 bp, respectively. The IS*L3* element was found in the upstream region of *fosA7.5* from isolates Kpg84, ECg85 (type I), and fEc.1 (type II), with lengths of 1,335 bp and 1,284 bp. Unlike the other two types, a sequence containing four IS*3*-

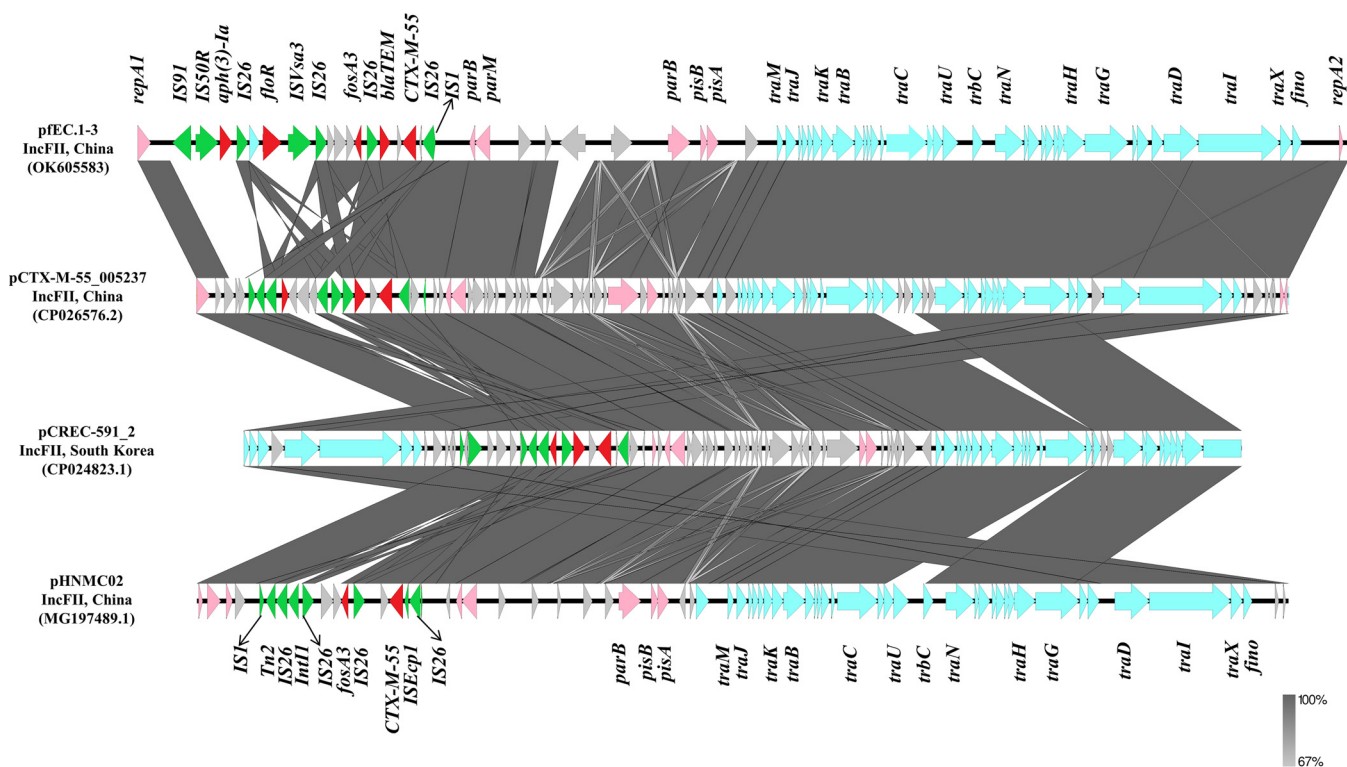

**FIG 7** Comparison of the genetic environment of *fosA3* in pfEc.1-3 and other closely related IncFII plasmids.

like elements (IS*911*, 303 bp; IS*EC52*, 657 bp; IS911, 303 bp; and IS*EC52*, 657 bp) was confirmed upstream of *fosA7.5* in type III. Also, the mobile elements of type III were highly similar to the IS*3* element, located downstream of *fosA7.5* in *E. coli* AH01, but the genetic direction is opposite (Fig. 9).

The *fosA7.5* gene from *E. coli* in this study was 100% identical to *fosA7.5*$^{WT}$ (wild-type *fosA7.5* sequence; WP_000941933.1), whereas it differed from the novel *fosA7* variant *fosA7.5*$^{Q86E}$ (EC623772). The antimicrobial susceptibility testing also confirmed that the 10 *fosA7.5*-positive *E. coli* isolates in this study showed high-level resistance to fosfomycin (MIC $\geq$ 512 $\mu$g/mL). In addition, fimbriae proteins (FimB, FimA, and FimE), bacterial membrane proteins (YijC and YijN), along with T3SS and T6SS secretion systems were identified on *fosA7.5*-bearing chromosomes in strains fEC.1 and fECg99-1, all of which were connected to bacterial virulence. Moreover, genes for the two-component regulatory systems, resistance, and efflux pumps related to antibiotic resistance were also found on the chromosomes, for example, genes for the two-component regulatory system Arls, PmcR, and PmrE and the *mcr-3* gene involved in colistin resistance, as well as the ARGs *bla*$_{CMY-63}$, *sul3*, *tetA*, and *dfrA* (Fig. 9).

**Phylogenetic analysis of fEC.99-1.** Similarly, the *fosA7.5*-carrying *E. coli* ST2599 strain fECg99-1 was studied by core genome MLST (cgMLST)-based phylogenomic analysis with 27 *E. coli* strains in GenBank (ST2599, *n* = 15; ST847, *n* = 10; ST6243, *n* = 1; ST4017, *n* = 1). ST2599 and ST847 have 6 identical alleles and differ only in their *adk* alleles. The results showed that the *E. coli* isolates from different parts of the world and multiple sources (human, cow, chicken, mouse, and pigeon) clustered together. Isolate fECg99-1 was found to be in the same lineage as ST2599 isolates from humans, in which two isolates 907357 (AXUH01) and A348 (NSAT01) collected from China were most similar to fECg99-1, indicating that the *E. coli* ST2599 strain has spread between humans and animals. Moreover, ST847 *E. coli* from Australia, the United States, India, and Mexico shared clonal similarities with fEC.99-1. In addition, all strains carried *fosA7* and also showed a multiresistance gene profile (Fig. 10).

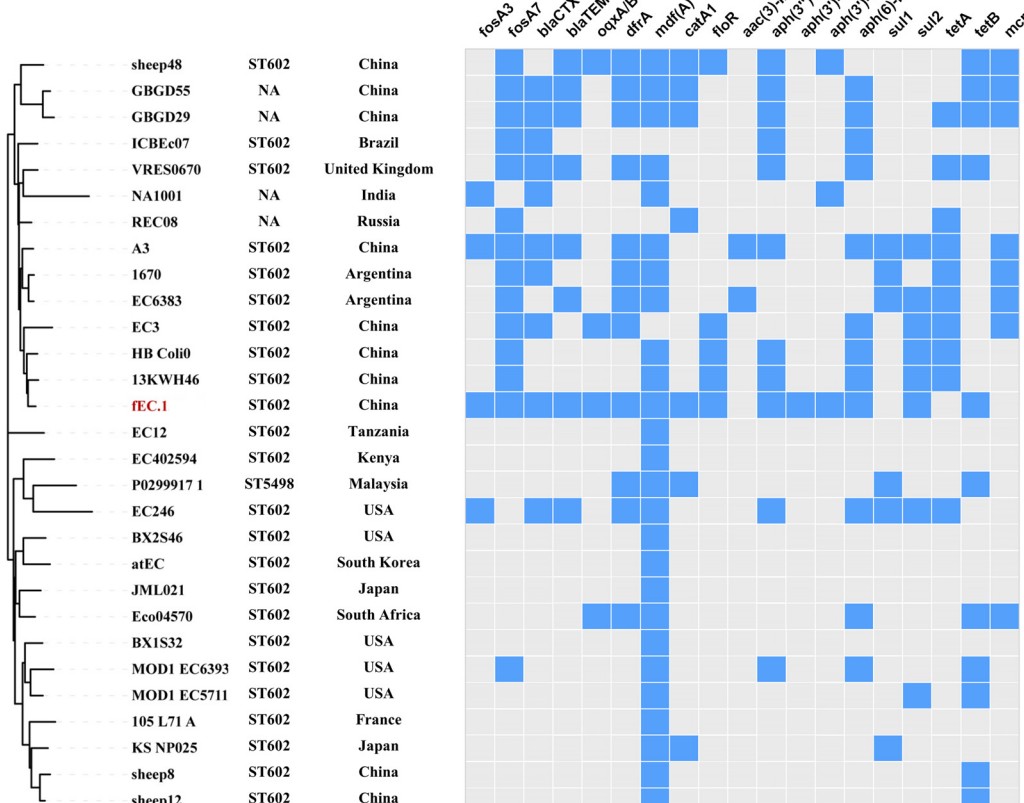

**FIG 8** Phylogenetic relationship of ST602 *E. coli* isolate fEC.1 (in red) from this study with ST602 isolates from China and other countries. Blue and gray squares indicate the presence and absence of antimicrobial resistance genes, respectively.

Finally, to determine whether *fosA7.5* in this study could confer resistance to fosfomycin, we created a recombinant plasmid, pET-28a+*fosA7.5* (Fig. S5). The fosfomycin MIC for *E. coli* Top10 transformed with pET-28a+*fosA7.5* was >128 $\mu$g/mL, which was more than 64-fold higher than that for *E. coli* Top10 transformed with pET-28a alone (Table 6).

## DISCUSSION

Fosfomycin has been used all over the world to treat clinical urinary tract infections. However, with the irregular use of antibiotics, the problem of fosfomycin resistance has gradually become serious. The use of fosfomycin in veterinary medicine has not been approved in China. However, this study revealed that the *fosA*-like genes in animal-derived *Enterobacteriaceae* isolates have a general prevalence, with *fosA3* (26%) having the highest detection rate. This rate was higher than the previously reported positivity rate for *fosA3* in humans, ducks, and pets (7, 10, 19). In addition, all strains containing *fosA*-like genes in this study exhibited a high-level resistance to fosfomycin (MIC $\geq$ 512 $\mu$g/mL).

According to previous reports (20, 21), *fosA* was frequently discovered on the chromosome of *E. cloacae* or on the transposon Tn*2921* of *S. marcescens*, while data on *fosA* prevalence are lacking. In this study, a total of 20 *fosA*-positive strains were identified, with a rate of 10%. A recent study reported the discovery of *fosA* in pet-derived *E. cloacae* from Taiwan, China, and similar to this study, 2 strains carried both *fosA* and *fosA3* (18). In addition, 10 *fosA7.5*-positive *E. coli* isolates were obtained from pigeons, and three of them also harbored *fosA3*. Since the identification of *fosA7* on the chromosomes of *Salmonella enterica* serovar Heidelberg from chickens in 2017, it has been detected in different sources, such as humans and birds (17, 22). In a previous study (23), it was found that all *fosA7*-positive *Salmonella* isolates were susceptible to fosfomycin, whereas *fosA7.5* detected in this study can confer high-level fosfomycin resistance (MIC $\geq$ 512 $\mu$g/mL) in *E. coli*. It is worth

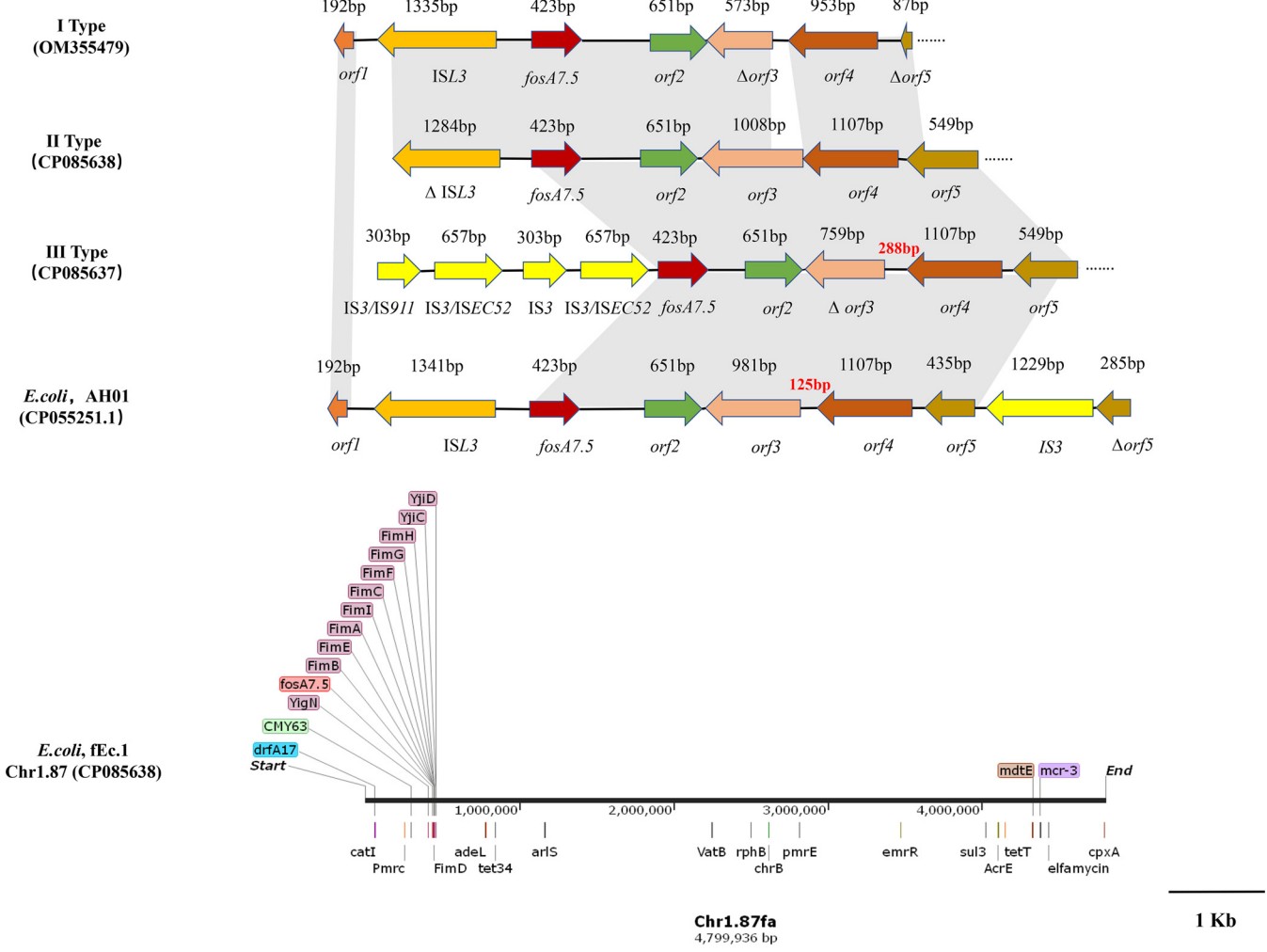

**FIG 9** Genetic context of *fosA7.5* in *E. coli*. Arrows indicate the directions of transcription of the genes, and different genes are shown in different colors. Shaded boxes between sequences indicate homologous regions (>90% sequence identity). The letter Δ indicates a truncated gene.

noting that as avians, pigeons can transmit the strains carrying the *fosA7*-like gene into other natural habitats, which seems to provide a pathway for the spread of resistance genes. The above analysis shows that *fosA7* gene has appeared in food animals, birds, and environments where humans live, and pigeons might be considered a source or vector of resistant isolates posing a threat to public and animal health.

In this study, except for one strain that was resistant to only two antibiotics, all *fosA/fosA3/fosA7.5*-bearing *Enterobacteriaceae* isolates were MDR and displayed a high rate of resistance to ceftazidime, florfenicol, tetracycline, and ciprofloxacin. The high prevalence of drug resistance in fosfomycin-resistant strains may be related to the overuse of these drugs in farms. Furthermore, we also found that *fosA3* or *fosA7.5* was often coharbored with *bla*$_{CTX-M}$, *floR*, and *bla*$_{TEM}$ in the same strain, similar to a previous report (24), which is likely to facilitate the dissemination and maintenance of *fosA3* by coselection. However, the current study identified only 9 *rmtB*-producing isolates (9/52), which was in contrast to a prior study (25). In China, because of the widespread use of tetracycline, cephalosporins, aminoglycosides, and florfenicol as treatments or feed additives in animal husbandry, strains containing *fosA*-like genes have a high occurrence of other resistance genes (26). Therefore, limiting the use of antibiotics in animal agriculture may help prevent the spread of *fosA*-like genes in strains.

In this study, ERIC-PCR typing showed 6 unique clusters and 11 ERIC types for 52 *fosA3*-carrying *E. coli* isolates, which revealed genetic diversity. Moreover, some isolates had

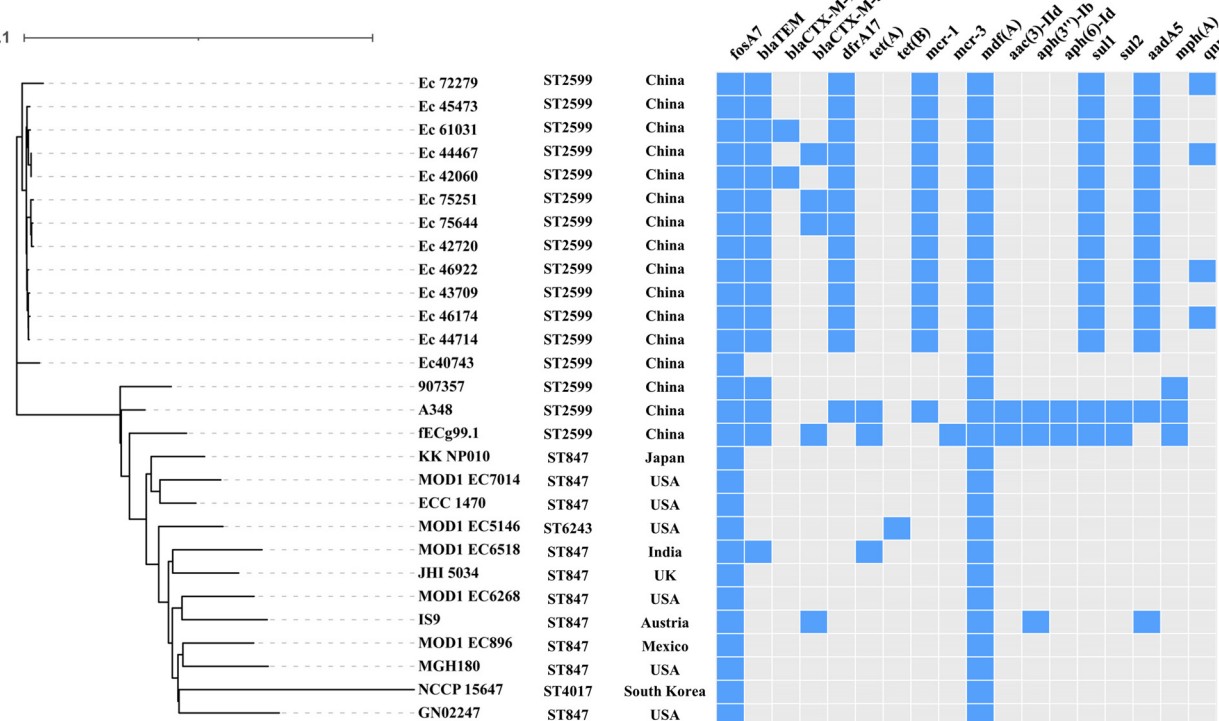

**FIG 10** Phylogenetic relationship of ST2599 *E. coli* isolate fECg99-1 from this study with isolates from China and other countries. Blue and gray squares indicate the presence and absence of antimicrobial resistance genes, respectively.

identical ERIC profiles, indicating dissemination from a similar origin. This result was consistent with a previous report that there was both clonal and horizontal transmission of these *fosA3*-positive *E. coli* (27). MLST analysis of 29 conjugable *fosA3*-positive *E. coli* isolates identified 15 STs, and ST115 was the most prevalent type, followed by ST156. However, ST115 and ST156 were previously found in ESBL-producing *E. coli* strains recovered from food and human samples (28, 29). MLST combined with ERIC-PCR analyses indicated that the 10 *fosA7.5*-positive *E. coli* isolates were mainly cloned among pigeons, which should arouse attention. At the same time, it demonstrated that the prevalence of fosfomycin-resistant strains has gradually increased, resulting in more serious problems of drug resistance.

The *fosA* gene was reported on conjugative plasmids or transposon Tn*2921* of *S. marcescens* strains (20, 21) in which the encoded protein FosA^Tn*2921* is closely related to FosA, encoded on the chromosome of *E. cloacae*, indicating that *fosA* has been trans-

**TABLE 6** MICs for constructed and original strains

| Antibiotic | MIC (μg/mL) for fEC.1 | | |
| --- | --- | --- | --- |
| | Alone | With pET-28a(+)-*fosA7.5*-Top10 | With pET-28a(+)-Top10 |
| Ceftazidime | 32 | <1 | <1 |
| Florfenicol | 512 | 4 | 2 |
| Chloramphenicol | 256 | 2 | 2 |
| Tetracycline | 32 | 8 | 4 |
| Ciprofloxacin | 128 | <1 | <1 |
| Amikacin | <1 | <1 | <1 |
| Colistin | <1 | <1 | <1 |
| Tigecycline | <0.25 | <0.25 | <0.25 |
| Meropenem | <1 | <1 | <1 |
| Ampicillin | >512 | 256 | 32 |
| Fosfomycin | >512 | >128 | 2 |

ferred between strains. All *fosA*-positive isolates in this study showed high levels of resistance to fosfomycin, but no *fosA*-carrying transconjugants were obtained, implying that *fosA* might be located on the chromosomes or nonconjugative plasmids of these *Enterobacteriaceae* isolates. Upon analysis of the genetic environment of *fosA*, a partial sequence similar to the transposon Tn*2921* and *E. cloacae* ECNIH5 was found, which suggested that mobile elements or transposons were the primary reason for the extensive spread of *fosA* among *Enterobacteriaceae*.

Our findings showed that *fosA3* was successfully transferred from donors to the recipient *E. coli* C600, implying that *fosA3* could be horizontally transferred to different bacterial individuals. Furthermore, this work identified six genetic environments of *fosA3*, and *fosA3* was frequently flanked by IS*26*, consistent with previous studies (30). Besides IS*26*, the different mobile elements identified in the regions surrounding *fosA3* and other resistance genes by WGS analysis include IS*91*, IS*4*, IS*Vsa3*, and IS*1*. These elements might play an important role in spreading antimicrobial resistance genes in Gram-negative bacteria (31). In short, the diversity of genetic contexts reflects the complexity of *fosA3* transmission in *E. coli*. According to previous reports, the *fosA3*-carrying plasmids were mainly IncFII, IncN, and IncFIB plasmids (32). In this study, *fosA3* was discovered on the conjugative IncFII plasmid. Additionally, the full sequence comparison analysis of plasmid showed that the IncFII plasmid in this study has high homology (>99%) with other IncFII plasmids carrying *fosA3* from different sources, especially humans and chickens, suggesting that *fosA3*-bearing IncFII plasmids are widely present in animals and humans.

Contrary to previous reports (17), no *fosA7.5*-carryig transconjugants were obtained in this study. The *fosA7.5* gene was located on the chromosomes of *E. coli* isolates belonging to ST602 and ST2599 and shared 100% similarity with *fosA7.5*<sup>WT</sup>. This study showed that *fosA7.5* could confer resistance to fosfomycin, because of the amino acid difference between FosA7.5 found in *E. coli* and FosA7 first found in *Salmonella* serovar Heidelberg, which is a crucial factor for the *fosA7.5* gene to show resistance to fosfomycin in *E. coli*. In this study, the isolates frequently contained insertion sequences (IS*L3* and IS*3*) both upstream and downstream of *fosA7.5*. As previously reported (33), *fosA7* alleles on the chromosomes could act as reservoirs of potential resistance genes, and they can be captured by mobile genetic elements to horizontally disseminate between different bacteria. In addition, *fosA7.5*-positive *E. coli* ST602 and ST2599 were found to be clonally transmitted, leading to an increased risk of drug resistance transmission to humans via the food chain, which could pose a serious threat to public health.

In conclusion, this study revealed a high prevalence and complex genetic environment of *fosA*-like genes in farm samples. Whether *fosA*-like genes are located on the chromosomes or plasmids of isolates, they may spread, mediated by mobile elements. The fosfomycin resistance gene is potentially transferred to the human body through the food chain, thus increasing the risk for human public health, and should be regularly monitored.

## MATERIALS AND METHODS

**Bacterial strains.** From September 2019 to December 2020, a total of 531 samples were collected from animals (chicken, pig, and pigeon) and their surroundings (sewage and soil) in farms in Guangxi Province, China. All samples were screened for the presence of fosfomycin-resistant isolates. Briefly, the samples were placed into LB broth and shaken at 37°C for approximately 16 to 18 h. Then, the fosfomycin-resistant isolates were selected on xylose-lysine-deoxycholate (XLD) agar plates (*Enterobacteriaceae* identification medium) containing 256 μg/mL fosfomycin. From each sample, only a single isolate of any one species was obtained. The strains were further identified using 16S rRNA sequencing (34), using primers described previously (F, AGAGTTTGATCATGGCTC; R, GGTTACCTTGTTACGACTT).

**Identification of fosfomycin-resistant determinants and the coexisting resistance genes.** The existence of fosfomycin-modifying-enzyme genes (*fosA3*, *fosA*, *fosC2*, *fosA7.5*, and others) in all selected fosfomycin-resistant isolates was determined by PCR and sequencing (18), and the *fosA7.5* primer was designed based on the sequence of *fosA7* (17). The surrounding regions of the *fosA*-like genes were determined by PCR mapping and sequencing using previously published primers (18, 24). Furthermore,

the florfenicol resistance gene *floR*, the 16S rRNA methyltransferase gene *rmtB*, the carbapenem resistance gene *bla*$_{NDM}$, the ESBL genes *bla*$_{CTX-M}$ (groups 1, 2, 8, and 9) and *bla*$_{TEM}$, and the plasmid-mediated AmpC lactamase gene *bla*$_{CMY-2}$ were also identified using PCR and sequencing (35–38). All primers are listed in the supplemental material.

**Antimicrobial susceptibility testing.** The MICs of 12 antibiotics (ceftazidime, florfenicol, chloramphenicol, erythromycin, tetracycline, ciprofloxacin, amikacin, meropenem, colistin, tigecycline, fosfomycin, and rifampicin) for the *fosA*-like gene-positive isolates were determined by the agar dilution method or broth microdilution method according to the CLSI (39). The MICs of fosfomycin were determined by the agar dilution method on Mueller-Hinton agar supplemented with 25 $\mu$g/mL glucose-6-phosphate (G-6-P), and the resistant breakpoints were recommended by the EUCAST in 2020 (40). *E. coli* ATCC 25922 was used as the control strain.

**Conjugation assays and plasmid replicon typing.** The transferability of fosfomycin resistance genes was determined by broth mating method using the plasmid-free *E. coli* C600 strain (Rif$^r$) as the recipient. Transconjugants were selected on MacConkey agar plates containing fosfomycin (100 $\mu$g/mL), G-6-P (25 $\mu$g/mL), and rifampicin (250 $\mu$g/mL) and finally confirmed by ERIC-PCR. When the conjugation experiments failed, *E. coli* DH5$\alpha$ was used as the recipient for transformation experiments. The transfer of the fosfomycin resistance genes (*fosA*, *fosA3*, or *fosA7.5*) was confirmed by PCR, and the MICs of transconjugants were also detected as described above. PCR-based replicon typing (PBRT) was used to screen the plasmid incompatibility groups for the *fosA*-like gene-positive isolates and their corresponding transconjugants (41). The primers are listed in the supplemental material.

**MLST and ERIC-PCR.** The 29 conjugable *fosA3*-positive *E. coli* isolates and the 10 *fosA7.5*-harboring isolates were subjected to MLST analysis, which was performed as previously described (42). The STs were obtained from the MLST database website (http://mlst.warwick.ac.uk/mlst/dbs/Ecoli). ERIC-PCR was carried out by using the primers ERIC-1 and ERIC-2 for *fosA3*-positive and *fosA7.5*-positive *E. coli* isolates (43). The isolated *Enterobacteriaceae* DNA samples were amplified in order to construct a computerized dendrogram, with the presence and absence of bands assumed to be 1 and 0, respectively. Following software processing, a matrix diagram of the binary number sequence was created and imported into NTSYS-pc (version 2.10) to perform the cluster analysis (44), which is based on the unweighted pair group method with arithmetic averages (UPGMA). Cluster were defined as being the same when the similarity between ERIC-PCR profiles was >80%.

**Whole-genome sequencing and phylogenetic analysis.** Whole-genome sequencing of two representative *E. coli* isolates (fEC.1 and fEC.99-1) from pigeons was performed. The extracted total genomic DNA of isolates was sequenced using the Nanopore PromethION and Illumina NovaSeq PE150 sequencing platforms, and the reads were assembled using Unicycler software. The coding sequences of the genetic context surrounding *fosA3* and *fosA7.5* were analyzed using the ORF Finder program (www.ncbi.nlm.nih.gov/gorf/orfig.cgi), and annotation was performed using the RAST server (http://rast.nmpdr.org/). The plasmid replicon types and antibiotic resistance genes prediction were analyzed using tools found at http://pubmlst.org/plasmid/ and https://cge.cbs.dtu.dk/services/. Genome comparison analysis of plasmids was performed using Easyfig and BRIG. WGS information for *E. coli* isolates was downloaded from GenBank (Tables S5 and S6), and cgMLST was performed as described previously (45).

**Cloning, expression, and functional verification of *fosA7.5*.** The *fosA7.5* gene from *E. coli* fEC.1 was cloned into pET-28a(+) and was transferred into *E. coli* Top10 by heat shock. Transformants were selected on LB agar plates containing 100 $\mu$g/mL kanamycin. Then, the recombinant clones were identified by PCR and Sanger sequencing. The Top10 strain containing pET-28a(+)-*fosA7.5* and the Top10 control strain were subjected to a fosfomycin resistance test to verify it's functionality.

**Data availability.** The *fosA7.5*-bearing chromosome sequences of fEc.1 and fEcg99-1 were submitted to NCBI with the accession numbers CP085638 and CP085637, respectively. The *fosA3*-bearing plasmid (pfEc.1-3) sequence was submitted with the accession number OK605583. The nucleotide sequences of *fosA* (types I, II, III, and IV), *fosA3* (types I, II, IV, V, and VI), and *fosA7.5* (type I) in this study have been deposited in GenBank under the accession numbers OM355477, OM289150, OM289151, OM355478, OM420281, OM355482, OM355480, OM355481, OM355483, and OM355479.

## SUPPLEMENTAL MATERIAL

Supplemental material is available online only.

**SUPPLEMENTAL FILE 1**, PDF file, 0.7 MB.

## ACKNOWLEDGMENTS

This work was supported by The Key Research and Development Plan of Guangxi, China (AB19245037), Natural National Science Foundation of China (31760746), and the Major R&D Project of Nanning Qingxiu District (2020005).

Xiaoxiao Zhang and Mingxiang Ma analyzed and interpreted the data. Yiqin Huang, Yajing Qian, Yuxiao Tan, Yujie Lu, Yumeng Cheng, and Xin Zhong performed the experiments and collected the data. Yunqiao Yang contributed to the revision of the article. Hongbin Si designed this work. All authors agreed on and approved the final manuscript.

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
