## [Reviewer comments · Microbiology Spectrum]

Microbiology Spectrum

Study on the Spread and Molecular Characteristics of Enterobacteriaceae Carrying fosA-like Genes from farms in China

xiaoxiao zhang, Mingxiang Ma, Yumeng Cheng, Yiqin Huang, Yuxiao Tan, Yunqiao Yang, Yajing Qian, Xin Zhong, Yujie Lu, and SI Hongbin

Corresponding Author(s): SI Hongbin, Guangxi University

Review Timeline:

Submission Date:	February 11, 2022
Editorial Decision:	April 12, 2022
Revision Received:	June 2, 2022
Accepted:	June 18, 2022

Editor: Thomas Denes

Reviewer(s): Disclosure of reviewer identity is with reference to reviewer comments included in decision letter(s). The following individuals involved in review of your submission have agreed to reveal their identity: Alexa R Cohn (Reviewer #2)

Transaction Report:

DOI: <https://doi.org/10.1128/spectrum.00545-22>

April 12, 2022

Prof. SI Hongbin
Guangxi University
No. 100, Daxuedong Road
Nanning
China

Re: Spectrum00545-22 (Study on the Spread and Molecular Characteristics of Enterobacteriaceae Carrying fosA-like Genes from farms in China)

Dear Prof. SI Hongbin:

While we are willing to consider a revised version of this paper at Spectrum, it would be in your best interest to improve the writing. I recommend that you ask a colleague of yours who is a native English speaker to read and provide you some feedback on the writing. You are also welcome to use one of the services here: <https://journals.asm.org/content/language-editing-services>

Link Not Available

Sincerely,

Thomas Denes

Journals Department
Reviewer comments:

Reviewer #1 (Comments for the Author):

The Authors wrote a manuscript on the spread and molecular characteristics of Enterobacteriaceae carrying fosA-like genes from farms in China. The manuscript is interesting and reports on quite original data. However, several issues should be addressed in order to increase its clarity and value:

Major issues

1. Lines 175-195. The *fosA* gene is resident in *Enterobacter cloacae* complex. It would be better to concentrate the attention in the manuscript only on the acquired genes *fosA3* and *fosA7*. I would suggest to remove data from the other parts of the text regarding *fosA*-positive *Enterobacter* spp as well.
2. Alterations in *glpT* and *uhpT* genes should be checked in the strains subjected to Whole Genome sequencing in order to exclude the presence of other mechanisms of resistance to fosfomycin.
3. Lines 455-458. Fosfomycin should be tested with agar dilution method on Mueller Hinton agar as recommended by ISO 20776-1:2019, CLSI and EUCAST.
4. Please add in the Discussion section a possible explanation of the high prevalence of acquired fosfomycin resistance genes in the veterinary setting and which measures should be taken in order to limit their spread.
5. Lines 312-350. ERIC PCR and MLST analysis sections should be combined and results reported together.
6. A revision of the use of English language is highly recommended.

Minor issues

1. Line 143. Enterobacteriaceae are intrinsically resistant to erythromycin. The evaluation of the antimicrobial activity of erythromycin adds limited information to the manuscript.
2. Line 90. "FOSA" protein should be written as "FosA".
3. Line 154. The mechanism of resistance to meropenem should be investigated, since carbapenem resistance is an important clinical issue.
3. Line 158. Please rephrase
4. Lines 369-370. 9 RmtB-producing isolate.
5. Please report all species and genes in italics.

Reviewer #2 (Comments for the Author):

In this study, Zhang et al. identified *fosA*-like genes in Enterobacteriaceae and the genetic context of such elements. The study shows that the identified *fosA*-like genes are frequently associated with mobile elements, which may suggest that these genes are highly transferable, allowing for their dissemination within microbial populations. Additionally, many of the isolates containing *fosA*-like elements also contained other antibiotic resistance genes. I recommend re-writing sections of the manuscript and have provided some examples below. I have a few suggestions for the authors to consider:

1. Move "(MDR)" on line 54 to after "multidrug-resistant" and before "Gram-negative bacteria"
2. Move "such as extended-spectrum- β -lactamase (ESBL)-producing Enterobacteriaceae and carbapenem-resistant Enterobacteriaceae (CRE)" after "has resulted in an increase in the number of bacterial infections" and before "while limiting the availability of antibiotics".
3. More information on *fosA* and *fosA*-like genes should be included in the introduction (i.e., clarification on which type of bacteria carry *fosA* genes, what the differences are between *fosA* and *fosA*-like genes).
4. The numbering of the tables included in the manuscript is off. The first table referenced in the manuscript should be titled "Table 1".
5. Please define "ARGs" on line 119.
6. Please add the antibiotics that the genes listed in lines 118-129 confer resistance to in parentheses.
7. Please define PLSDB on line 232.
8. Please differentiate between supplemental tables and supplemental datasets.
9. Please elaborate more on the correlation between bacterial resistance and virulence mentioned in the discussion.
10. Were any quality checks performed on the assembled sequences (i.e., those isolates containing *fosA7.5*)?

Reviewer #3 (Comments for the Author):

This manuscript presents the analysis of isolates from farm samples of Enterobacteriaceae but focusses mostly on *E. coli* isolates that are Fosfomycin resistant. The authors perform phenotypic testing of antimicrobial resistance and genetic analysis of the genes around the fosfomycin resistance genes. They identify transposon fragments and plasmids which encode these genes, however most of the transposons are not transferable in vitro. The authors also look at the genetics of the isolates through ERIC PCR and four are subjected to WGS analysis. A major issue with the paper is improper English usage. While most of the paper is grammatically correct, the words used or the way they are used do not make sense or are confusing as to what the authors are trying to communicate. The paper could benefit from editing by a native English speaker or an English editorial service. In addition to the confusion caused by this, the authors could also consider some of these points:

1. English usage
2. L.21-25, What does concomitant mean in this sentence?
3. L.26, "successfully obtained" is confusing usage
4. L.42, connected is the wrong word
5. L.44-46, this section is confusing to the reader. You list 10 *FosA7.5* containing isolates but then discuss on three. I think you are only discussing the three that are *E. coli*. Is that correct? Please rewrite the sentence to make it clear to the reader.

6. L47-49, explaining that this work is important because it is important to complete the work is a circular argument. Why is completing the work important, what will that work inform us of?
7. L70, By America, do you mean the USA, or North America or South America or both? If the whole hemisphere, please refer to it as, "the Americas"
8. L84, it is not co-localized, please rephrase.
9. L128, This is not correct, they do not coexist, rather they are both in the same isolate.
10. L131, Table 3? Where are Table 1 and 2? Please reorganize the order of the tables and figures to follow the text.
11. L391 Awkward sentence, please rewrite it to make it clear to the reader what you mean.
12. L397, Awkward sentence, please rewrite.
13. L412, "verified in THIS study"? I think you mean it should be verified in a subsequent study.

Staff Comments:

Preparing Revision Guidelines

Please return the manuscript within 60 days; if you cannot complete the modification within this time period, please contact me. If you do not wish to modify the manuscript and prefer to submit it to another journal, please notify me of your decision immediately so that the manuscript may be formally withdrawn from consideration by Microbiology Spectrum.

While we are willing to consider a revised version of this paper at Spectrum, it would be in your best interest to improve the writing. I recommend that you ask a colleague of yours who is a native English speaker to read and provide you some feedback on the writing. You are also welcome to use one of the services here: <https://journals.asm.org/content/language-editing-services>

While we are willing to consider a revised version of this paper at Spectrum, it would be in your best interest to improve the writing. I recommend that you ask a colleague of yours who is a native English speaker to read and provide you some feedback on the writing. You are also welcome to use one of the services here: <https://journals.asm.org/content/language-editing-services>

In this study, Zhang et al. identified *fosA*-like genes in Enterobacteriaceae and the genetic context of such elements. The study shows that the identified *fosA*-like genes are frequently associated with mobile elements, which may suggest that these genes are highly transferable, allowing for their dissemination within microbial populations. Additionally, many of the isolates containing *fosA*-like elements also contained other antibiotic resistance genes. I recommend re-writing sections of the manuscript and have provided some examples below. I have a few suggestions for the authors to consider:

1. Move “(MDR)” on line 54 to after “multidrug-resistant” and before “Gram-negative bacteria”
2. Move “such as extended-spectrum- β -lactamase (ESBL)-producing *Enterobacteriaceae* and carbapenem-resistant *Enterobacteriaceae* (CRE)” after “has resulted in an increase in the number of bacterial infections” and before “while limiting the availability of antibiotics”.
3. More information on *fosA* and *fosA*-like genes should be included in the introduction (i.e., clarification on which type of bacteria carry *fosA* genes, what the differences are between *fosA* and *fosA*-like genes).
4. The numbering of the tables included in the manuscript is off. The first table referenced in the manuscript should be titled “Table 1”.
5. Please define “ARGs” on line 119.
6. Please add the antibiotics that the genes listed in lines 118-129 confer resistance to in parentheses.
7. Please define PLSDB on line 232.
8. Please differentiate between supplemental tables and supplemental datasets.
9. Please elaborate more on the correlation between bacterial resistance and virulence mentioned in the discussion.
10. Were any quality checks performed on the assembled sequences (i.e., those isolates containing *fosA7.5*)?

Response to Reviewers

(Modifications in the manuscript are highlighted in yellow)

Questions:

Reviewer 1

Major issues

1. Lines 175-195. The *fosA* gene is resident in *Enterobacter cloacae* complex. It would be better to concentrate the attention in the manuscript only on the acquired genes *fosA3* and *fosA7*. I would suggest to remove data from the other parts of the text regarding *fosA*-positive *Enterobacter spp* as well.
2. Alterations in *glpT* and *uhpT* genes should be checked in the strains subjected to Whole Genome sequencing in order to exclude the presence of other mechanisms of resistance to fosfomycin.
3. Lines 455-458. Fosfomycin should be tested with agar dilution method on Mueller Hinton agar as recommended by ISO 20776-1:2019, CLSI and EUCAST.
4. Please add in the Discussion section a possible explanation of the high prevalence of acquired fosfomycin resistance genes in the veterinary setting and which measures should be taken in order to limit their spread.
5. Lines 312-350. ERIC PCR and MLST analysis sections should be combined and results reported together.
6. A revision of the use of English language is highly recommended.

Minor issues

1. Line 143. Enterobacteriaceae are intrinsically resistant to erythromycin. The evaluation of the antimicrobial activity of erythromycin adds limited information to the manuscript.
2. Line 90. "FOSA" protein should be written as "FosA".
3. Line 154. The mechanism of resistance to meropenem should be investigated, since carbapenem resistance is an important clinical issue.
4. Line 158. Please rephrase.
5. Lines 369-370, 9 rmtB-producing isolates.
6. Please report all species and genes in italics.

Reviewer 2

1. Move "(MDR)" on line 54 to after "multidrug-resistant" and before "Gram-negative bacteria".
2. Move "such as extended-spectrum- β -lactamase (EBSL)-producing Enterobacteriaceae and carbapenem-resistant *Enterobacteriaceae* (CRE)" after "has resulted in an increase in the number of bacterial infections" and before "while limiting the availability of antibiotics".
3. More information on *fosA* and *fosA*-like genes should be included in the introduction (i.e., clarification on which type of bacteria carry *fosA* genes, what the differences are between *fosA* and *fosA*-like genes).
4. The numbering of the tables included in the manuscript is off. The first table referenced in the manuscript should be titled "Table 1".
5. Please define "ARGs" on line 119.
6. Please add the antibiotics that the genes listed in lines 118-129 confer resistance to in parentheses.
7. Please define PLSDB on line 232.
8. Please differentiate between supplemental tables and supplemental datasets.
9. Please elaborate more on the correlation between bacterial resistance and virulence mentioned in the discussion.
10. Were any quality checks performed on the assembled sequences (i.e., those isolates containing *fosA7.5*)?

Reviewer 3

1. English usage.
2. L.21-25, what does concomitant mean in this sentence?
3. L.26, "successfully obtained" is confusing usage.
4. L.42, connected is the wrong word.
5. L.44-46, this section is confusing to the reader. You list 10 FosA7.5 containing isolates but then discuss on three. I think you are only discussing the three that are *E. coli*. Is that correct? Please rewrite the sentence to make it clear to the reader.
6. L47-49, explaining that this work is important because it is important to complete the work is a circular argument. Why is completing the work important, what will that work inform us of?
7. L70, By America, do you mean the USA, or North America or South America or both? If the whole hemisphere, please refer to it as, "the Americas".
8. L84, it is not co-localized, please rephrase.

9. L128, this is not correct, they do not coexist, rather they are both in the same isolate.
10. L131, Table 3? Where are Table 1 and 2? Please reorganize the order of the tables and figures to follow the text.
11. L391 Awkward sentence, please rewrite it to make it clear to the reader what you mean.
12. L397, Awkward sentence, please rewrite.
13. L412, "verified in THIS study"? I think you mean it should be verified in a subsequent study.

Response to Reviewers

Reviewer 1

Major issues

1. Lines 175-195. The *fosA* gene is resident in *Enterobacter cloacae* complex. It would be better to concentrate the attention in the manuscript only on the acquired genes *fosA3* and *fosA7*. I would suggest to remove data from the other parts of the text regarding *fosA*-positive *Enterobacter* spp as well.

As suggested by the reviewer, the genetic environment of *fosA* has been re-analyzed, and focused only on *E. cloacae* and *E. hormaechei* isolates. Modifications can be seen on lines 235-248. And the data related to ERIC-PCR typing of *fosA*-positive strains were removed. According to the reviewer's advice, in-depth analysis of the *fosA3* and *fosA7.5* positive strains was carried out, and the phylogenetic analysis of ST602 and ST2599 *E. coli* carrying either *fosA3* or *fosA7.5* with known full genome sequences was added to the manuscript, in order to understand the epidemic and transmission characteristics of these strains. Modifications can be seen on lines 283-297 and 331-343.

2. Alterations in *glpT* and *uhpT* genes should be checked in the strains subjected to Whole Genome sequencing in order to exclude the presence of other mechanisms of resistance to fosfomycin.

As suggested by the reviewer, the chromosome-mediated fosfomycin resistance mechanism of the two strains sequenced by the whole genome was analyzed, and it was found that the *uhpT* gene sequences of strains fECg99.1 and fEC.1 were identical to the reference strain K12. There were two amino acids were changed in GlpT of fEC.1, and some sequences were missing. While the target enzyme MurA was not

changed in the two strains. However, in strain fEC.1, *fosA3* is located on a conjugative IncFII plasmid. We verified by conjugation test that after obtaining *fosA3*-carrying IncFII plasmid, the MICs of fosfomycin to the recipient bacteria was $\geq 512\mu\text{g/mL}$. In strain fECg99.1, we tested the function of the *fosA7.5* gene, and the results showed that the *fosA7.5* gene identified in this study can confer high-level resistance to fosfomycin ($\text{MIC} > 128\mu\text{g/mL}$).

3. Lines 455-458. Fosfomycin should be tested with agar dilution method on Mueller Hinton agar as recommended by ISO 20776-1:2019, CLSI and EUCAST.

As suggested by the reviewer, the content of "Antimicrobial susceptibility testing" has been modified, and the modifications can be seen on lines 479-482.

4. Please add in the Discussion section a possible explanation of the high prevalence of acquired fosfomycin resistance genes in the veterinary setting and which measures should be taken in order to limit their spread.

As suggested by the reviewer, this aspect has been added to the "discussion". Modifications can be seen on lines 380-390. In addition, according to the suggestions of reviewers, the "discussion" section of the manuscript has been modified to provide an in-depth discussion of the findings, and the modifications can be seen on lines 351-443.

5. Lines 312-350. ERIC PCR and MLST analysis sections should be combined and results reported together.

As suggested by the reviewer, the ERIC PCR and MLST analysis sections have been combined, and the results also reported together. Modifications can be seen on lines 203-234.

6. A revision of the use of English language is highly recommended.

As suggested by the reviewer, the use of English language throughout the article has been revised.

Minor issues

1. Line 143. Enterobacteriaceae are intrinsically resistant to erythromycin. The evaluation of the antimicrobial activity of erythromycin adds limited information to the manuscript.

As suggested by the reviewer, the evaluation of the antibacterial activity of erythromycin and rifampicin in "Detection of Antimicrobial Resistance Patterns" has been removed, and the analysis of the results has been revised in detail. Modifications can be seen on lines 155-172.

2. Line 90. "FOSA" protein should be written as "FosA".

As suggested by the reviewer, the "FOS" that appears throughout the article has been changed to "Fos".

3. Line 154. The mechanism of resistance to meropenem should be investigated, since carbapenem resistance is an important clinical issue.

As suggested by the reviewer, the resistance mechanism of these strains to meropenem has been analyzed, and the results showed that meropenem-resistant strains carried *bla*_{NDM} resistance gene. Modifications can be seen on lines 129-131 and Table 2.

4. Line 158. Please rephrase.

As suggested by the reviewer, the sentence on line 158 has been restated and changed to "Except for one strain that was only resistant to two antibiotics (fosfomycin and florfenicol), all 81 strains carried *fosA*-like genes were multidrug-resistant strains (resistant to at least 3 classes agents)". Modifications can be seen on lines 168-170.

5. Lines 369-370. 9 RmtB-producing isolate.

As suggested by the reviewer, the sentence on lines 369-370 has been modified, and the modifications can be seen on lines 385-386.

6. Please report all species and genes in italics.

As suggested by the reviewer, the all species and genes in the text have been changed to italic status.

Reviewer 2

1. Move "(MDR)" on line 54 to after "multidrug-resistant" and before "Gram-negative bacteria".

As suggested by the reviewer, the content in the "Introduction" has been revised, and the modifications can be seen on lines 58-61.

2. Move "such as extended-spectrum-β-lactamase (EBSL)-producing Enterobacteriaceae and carbapenem-resistant Enterobacteriaceae (CRE)" after "has resulted in an increase in the number of bacterial infections" and before "while limiting the availability of antibiotics".

As suggested by the reviewer, the content in the "Introduction" has been revised, and the modifications can be seen on lines 58-61.

3. More information on *fosA* and *fosA*-like genes should be included in the introduction (i.e., clarification on which type of bacteria carry *fosA* genes, what the differences are between *fosA* and *fosA*-like genes).

As suggested by the reviewer, the content in the "Introduction" has been revised,

which on lines 73 and 88-90. The “*fosA*-like genes” represent all genes that have been identified to encode FosA enzymes, including *fosA1-fosA10* et al. The “*fosA* gene” that appears in the manuscript represents the “*fosA1* gene”. As the researchers used to describe the gene as “*fosA*” in previous research reports, this paper also describes “*fosA1*” as “*fosA*”.

4. The numbering of the tables included in the manuscript is off. The first table referenced in the manuscript should be titled "Table 1".

As suggested by the reviewer, the tables and figures appearing in the manuscript have been presented in order.

5. Please define “ARGs” on line 119.

As suggested by the reviewer, we have defined “ARGs” as “antibiotic resistance genes (ARGs)” on line 116.

6. Please add the antibiotics that the genes listed in lines 118-129 confer resistance to in parentheses.

As suggested by the reviewer, we have added the antibiotics that the genes confer resistance to in parentheses. And the modifications can be seen on lines 465-472, including fosfomycin-modifying enzyme genes (*fosA3*, *fosA*, *fosC2*, *fosA7.5*, and et al), florfenicol resistance gene *floR*, 16S rRNA methyltransferase gene *rmtB*, carbapenem-resistance gene *bla_{NDM}*, the ESBL genes *bla_{CTX-M}* (group 1, 2, 8, 9) and *bla_{TEM}*, and the plasmid-mediated AmpC-lactamase gene *bla_{CMY-2}*.

7. Please define PLSDB on line 232.

Since the BLAST method was used to screen out plasmids with higher similarity, the method of PLSDB was deleted. The modifications can be seen on lines 272-274.

8. Please differentiate between supplemental tables and supplemental datasets.

As suggested by the reviewer, the Supplementary Tables and Supplementary Figures have been added to the “Supplementary Materials”.

9. Please elaborate more on the correlation between bacterial resistance and virulence mentioned in the discussion.

Since the results of this study showed that there were multiple virulence-related factors around *fosA7.5*, but the relationship between the resistance gene and virulence was not confirmed, this section was deleted from the “Discussion”.

10. Were any quality checks performed on the assembled sequences (i.e., those isolates containing *fosA7.5*)?

The whole genome sequences, which we obtained were quality-checked.

Reviewer 3

1. English usage.

As suggested by the reviewer, the use of English language throughout the article has been revised.

2. L.21-25, what does concomitant mean in this sentence?

"Concomitant with" in the text means that three *fosA7.5*-carrying *E. coli* strains were also identified to carry the *fosA3* gene.

3. L.26, "successfully obtained" is confusing usage.

As suggested by the reviewer, "successfully obtained" has been changed to "was successfully transferred" on line 27.

4. L.42, connected is the wrong word.

As suggested by the reviewer, the "Importance" content in the "Abstract" has been revised. The modifications can be seen on lines 44-54.

5. L.44-46, this section is confusing to the reader. You list 10 FosA7.5 containing isolates but then discuss on three. I think you are only discussing the three that are E. coli. Is that correct? Please rewrite the sentence to make it clear to the reader.

As suggested by the reviewer, the incorrect expression in the "importance" section of the "Abstract" has been revised. The modifications can be seen on lines 44-54.

6. L47-49, explaining that this work is important because it is important to complete the work is a circular argument. Why is completing the work important, what will that work inform us of?

As suggested by the reviewer, the importance of this work has been described in "importance". The work indicated that the food animals served as a potential reservoir for the resistance genes, and the mobile elements would accelerate the transmission of *fosA*-like genes in strains. In addition, the *E. coli* carrying *fosA*-like genes from food animals had a high similarity with *E. coli* isolates from humans, suggesting that *fosA*-like genes can be transmitted to humans through the food chain, thus posing a serious threat to public health. The modifications can be seen on lines 46-53.

7. L70, By America, do you mean the USA, or North America or South America or both? If the whole hemisphere, please refer to it as, "the Americas".

As suggested by the reviewer, "America" in the manuscript has been changed to "the Americas", and is on line 69.

8. L84, it is not co-localized, please rephrase.

As suggested by the reviewer, this part of the "Introduction" has been re-described, and the modifications can be seen on lines 77-80.

9. L128, this is not correct, they do not coexist, rather they are both in the same isolate.

As suggested by the reviewer, this part of the "Identification of fosfomycin-resistant determinants and the coexistent-resistant genes" has been re-described, and the modifications can be seen on lines 126-128.

10. L131, Table 3? Where are Table 1 and 2? Please reorganize the order of the tables and figures to follow the text.

As suggested by the reviewer, the tables and figures appearing in the manuscript have been presented in order.

11. L391 Awkward sentence, please rewrite it to make it clear to the reader what you mean.

As suggested by the reviewer, this sentence has been revised, and the modifications can be seen on lines 412-413.

12. L397, Awkward sentence, please rewrite.

As suggested by the reviewer, this sentence has been revised, and the modifications can be seen on lines 418-421.

13. L412, "verified in THIS study"? I think you mean it should be verified in a subsequent study.

As suggested by the reviewer, the whole part of the "discussion has been revised".

In addition to this, each corresponding figure has been modified according to the revision of the content in the manuscript.

June 18, 2022

Prof. SI Hongbin
Guangxi University
No. 100, Daxuedong Road
Nanning
China

Re: Spectrum00545-22R1 (Study on the Spread and Molecular Characteristics of Enterobacteriaceae Carrying fosA-like Genes from farms in China)

Dear Prof. SI Hongbin:

Your manuscript has been accepted, and I am forwarding it to the ASM Journals Department for publication. You will be notified when your proofs are ready to be viewed.

Sincerely,

Thomas Denes
Editor, Microbiology Spectrum
